# ENHANCING OFFLINE REINFORCEMENT LEARNING WITH AN OPTIMAL SUPPORTED DATASET

## ABSTRACT

Offline Reinforcement Learning (Offline RL) is challenged by distributional shift and value overestimation, which often leads to poor performance. To address this issue, a popular class of methods use behavior regularization to constrain the learned policy to stay close to the behavior policy. However, this approach can be too limiting when the behavior policy is suboptimal. To overcome this limitation, we propose to conduct behavior regularization directly on an optimal supported dataset, which can both ensure that the learned policy is not too far removed from the dataset, and reduce any potential bias towards the optimization objective. We introduce *Optimal Supported Dataset generation via Stationary DIstribution Correction Estimation* (OSD-DICE) to generate such a dataset. OSD-DICE is based on the primal-dual formulation of linear programming for RL. It uses a single minimization objective to avoid poor convergence issues often associated with this formulation, and incorporates two key designs to ensure polynomial sample complexity under general function approximation and single-policy concentrability. After generating the near-optimal supported dataset, we instantiate our framework by two representative behavior regularization-based methods and show safe policy improvement over the near-optimal supported policy. Empirical results validate the efficacy of OSD-DICE on tabular tasks and demonstrate remarkable performance gains of the proposed framework on D4RL benchmarks.

## 1 INTRODUCTION

Offline Reinforcement Learning (RL) allows the policy to be learned from a fixed dataset without further interactions. However, the offline learning paradigm usually raises the distributional shift between the learning policy and the dataset, and thus suffers from severe value overestimation (Levine et al., 2020; Fujimoto et al., 2019). A major solution to this issue is the use of pessimism principle, which resorts to pessimistic value estimates to eliminate the negative impact of overestimation (Buckman et al., 2020; Jin et al., 2021; Xie et al., 2021; Cheng et al.). One broad category of practical methods fulfills pessimism through behavior regularization, which constrains the learned policy to lie close to the behavior policy. This is typically achieved by adding a penalty term to the critic or actor loss measuring the divergence of the learning policy from the behavior policy.(Kumar et al., 2019; Fujimoto et al., 2019; Wu et al., 2019; Kostrikov et al., 2021; Kumar et al., 2020; Fujimoto & Gu, 2021). Clearly, regularized optimization that is subject to behavior regularization typically leads to a sub-optimal solution, as it trades off the optimal policy against the behavior policy. And it is commonly known to be too restrictive to achieve good performance both theoretically and empirically given a poor behavior policy (Kumar et al., 2019; Wu et al.)

Actually, as pessimism principle indicates, the purpose of the behavior regularization is to act as a pessimistic penalty that ensures the learned policy is in support, as opposed to forcing the solution towards an improper behavioral policy. Therefore, any policy supported by the dataset can be regarded as the "behavior policy", and there is likely room for improvement by adopting a more effective "behavior policy". Ideally, the optimal policy supported by the dataset should serve as the preferred "behavior policy", as it both encourages the learned policy to stay in support and reduces any potential bias towards the optimization objective, compared to the original behavior policy. Based on this concept, we propose to implement behavior regularization with the optimal supported (in-support) policy in this paper. This can be achieved by conducting offline RL methods directly on the optimal supported dataset. Intuitively, the new dataset can be seen as having been collected according to

the optimal supported policy, and is expected to enjoy a much higher performance compared to the previous one. By doing so, the learned policy can be driven towards the optimal direction supported by the dataset.

The primal-dual formulation of Linear Programming (LP) for MDPs is suitable for obtaining the optimal supported dataset. In this approach, conventional policy optimization is transformed into a constrained (regularized) LP optimization, with distribution correction (or density ratio) treated as a variable. By reweighing the original dataset with the learned optimal density ratio, the intended dataset can be obtained spontaneously. Although significant theoretical progress has been made recently in this area (Zhan et al., 2022; Rashidinejad et al., 2022; Ozdaglar et al., 2022), the practical implementation of this formulation continues to be a challenge. They have typically relied either on multi-level nested optimization (Zhan et al., 2022; Rashidinejad et al., 2022), which can introduce well-known issues with numerical instability and local convergence (Goodfellow et al., 2020), or additional constraints that can be challenging to accommodate in practice (Ozdaglar et al., 2022). A related practical approach Lee et al. (2021) attempts to circumvent these limitations by transforming the multi-level optimization into a single minimization. However, this method is known to generate a biased objective on stochastic MDPs, due to the double-sampling design and nonlinear properties involved, making the resulting solution lack optimality guarantees.

In order to obtain a practical solution that is also backed up by strong theoretical support, we develop *Optimal Supported Dataset generation via Stationary DIstribution Correction Estimation* (OSD-DICE), which enjoys both practical advantage by adopting the single minimization objective, and optimality guarantee by the introduction of two key designs. To be more precise, we leverage the maximum likelihood of the transition model for advantage estimation instead of relying on single transition estimation, then the overall objective bias can be bounded small enough. Furthermore, we enhance the convexity of the objective function by introducing a squared regularization term. This term does not alter the optimal solution, but greatly benefits the smoothness of the objective. Building upon these improvements, we are able to establish polynomial sample complexity guarantees for OSD-DICE under general function approximation and single-policy concentrability.

After obtaining the near-optimal dataset, we instantiate our proposed framework in conjunction with two representative behavior regularization-based methods: Behavior Cloning (BC) and Conservative Q-Learning (CQL)(Kumar et al., 2020). This leads to optimal supported dataset-based BC (osd-BC) and optimal supported dataset-based CQL (osd-CQL) respectively. Additionally, we present the safe policy improvement over the near-optimal supported policy for both osd-BC and osd-CQL. Extensive experimental study is also conducted on tabular tasks and standard offline RL benchmark D4RL (Fu et al., 2020), which confirms the efficacy of OSD-DICE and showcases the performance enhancement of offline RL with the aid of an optimal supported dataset.

## 2   RELATED WORK

RL algorithms are particularly prone to failure when used in the offline setting, due to the erroneous value estimation induced by the distributional shift between the dataset and the learning policy. To address this issue, one broad category of methods adopts behavior regularization, which constrains the learned policy to stay close to the behavior policy, either explicitly or implicitly. Various implementations have been proposed, primarily differing in the choice of behavior regularizer, such as KL (Fujimoto et al., 2019; Wu et al., 2019), MMD (Kumar et al., 2019), and others (Kumar et al., 2020; Kostrikov et al., 2021; Fujimoto & Gu, 2021). This class of methods is conceptually supported by the pessimism principle, which suggests that pessimism can be incorporated into the policy evaluation process to mitigate overestimation and achieve good performance even with imperfect data coverage (Buckman et al., 2020; Jin et al., 2021; Liu et al., 2020b; Kumar et al., 2021; Rashidinejad et al., 2021; Xie et al., 2021; Zanette et al., 2021; Cheng et al.).

Some methods exploit the primal-dual formulation of LP instead of typical dynamic programming to learn(Lee et al., 2021; Zhan et al., 2022; Rashidinejad et al., 2022; Ozdaglar et al., 2022). These approaches learn optimal density ratio and then extract the optimal policy through the learned density ratio. This concept gains theoretical and empirical success in the off-policy evaluation (OPE) (Nachum et al., 2019a;b; Zhang et al., 2020; Nachum & Dai, 2020), and attains theoretical progress in offline RL field recently (Zhan et al., 2022; Rashidinejad et al., 2022; Ozdaglar et al., 2022). Although these methods have shown theoretical success, they present practical difficulties. For

example, (Zhan et al., 2022; Rashidinejad et al., 2022) involve multi-level nested optimization known to cause numerical instability and local convergence problems, while (Ozdaglar et al., 2022) rely on impractical extra constraints. While (Lee et al., 2021) attempts to make this formulation practical, it results in a biased objective function and lacks theoretical guarantees. Our proposed OSD-DICE builds on this by integrating two crucial designs into the single minimization objective introduced in (Lee et al., 2021), resulting in a practical algorithm with polynomial sample complexity.

There is a type of research that still exists which utilizes a reweighing strategy to reduce weights for out-of-distribution (OOD) data (Wu et al., 2021) or focus more on high return state-action pairs. In the latter category, one direct approach is to select the trajectory with the highest return in the dataset for subsequent learning (Emmons et al., 2021), or to imitate the action with the highest value function (Chen et al., 2020). The more common practice is to adopt exponentiated advantage or return estimates as importance weights, with different methods of advantage estimation(Wang et al., 2018; Liu et al., 2020a; Hong et al.; Yue et al., 2023; Xu et al., 2023). While simple and intuitive, most of these methods do not have a theoretical guarantee of obtaining the near-optimal supported policy. The most pertinent study to our research is Hong et al. (2023) conducted during the same period. This study also suggests utilizing optimal density ratios as weights.However, it is worth mentioning that Hong et al. (2023) primarily focuses on the scenario where $\gamma = 1$. Consequently, they propose a unique algorithm to estimate the optimal density ratio, but no evidence is provided to support its optimality. Our method targets the more common case of $\gamma < 1$ and theoretically guarantees learning the near-optimal density ratio. Additionally, when combined with typical offline RL algorithms, it can achieve near-optimal supported policies, which is not possessed by the aforementioned works.

## 3 BACKGROUND

**Markov decision process.** An infinite-horizon discounted MDP is described by a tuple $\mathcal{M} = (\mathcal{S}, \mathcal{A}, P, R, \rho, \gamma)$, where $\mathcal{S}$ is the state space, $\mathcal{A}$ is the action space, $P : \mathcal{S} \times \mathcal{A} \to \Delta(\mathcal{S})$ is the transition kernel, $R : \mathcal{S} \times \mathcal{A} \to \Delta([0, 1])$ encodes a family of reward distributions with $r : \mathcal{S} \times \mathcal{A} \to [0, 1]$ as the expected reward function, $\rho : \mathcal{S} \to \Delta(\mathcal{S})$ is the initial state distribution and $\gamma \in [0, 1)$ is the discount factor. We assume $\mathcal{S}$ and $\mathcal{A}$ are both finite sets. A stationary(stochastic) policy $\pi : \mathcal{S} \to \Delta(\mathcal{A})$ specifies a distribution over actions in each state. Each policy $\pi$ induces a discounted stationary distribution over state-action pairs $d^\pi : \mathcal{S} \times \mathcal{A} \to [0, 1]$ defined as $d^\pi(s, a) := (1 - \gamma) \sum_{t=0}^{\infty} \gamma^t P_t(s_t = s, a_t = a; \pi)$, where $P_t(s_t = s, a_t = a; \pi)$ denotes $(s, a)$ visitation probability at step $t$, starting at $s_0 \sim \rho(\cdot)$ and following $\pi$. We abuse notation and also write $d^\pi(s) = \sum_{a \in \mathcal{A}} d^\pi(s, a)$ to denote the discounted state stationary distribution.

An important quantity is the value of a policy $\pi$, which is the discounted sum of rewards $V^\pi(s) := \mathbb{E}\left[\sum_{t=0}^{\infty} \gamma^t r_t | s_0 = s, a_t \sim \pi(\cdot|s_t), \forall t \geq 0\right]$ starting at $s \in \mathcal{S}$. Q function $Q^\pi(s, a)$ of a policy is similarly defined. We write $J(\pi) := (1 - \gamma)\mathbb{E}_{s \sim \rho}[V^\pi(s)] = \mathbb{E}_{s,a \sim d^\pi}[r(s, a)]$ to represent a scalar summary of the performance of a policy $\pi$. We also denote by $\pi^*$ an optimal policy that maximizes the above objective.

**Offline reinforcement learning.** We focus on the offline RL, where the agent is only provided with a previously-collected offline dataset $\mathcal{D} = \{(s_i, a_i, r_i, s_i')\}_{i=1}^{N}$. Here, $r_i \sim R(s_i, a_i), s_i' \sim P(\cdot|s_i, a_i)$, and we assume $(s_i, a_i)$ pairs are independently and identically distributed(i.i.d.) according to a data distribution $\mu \in \Delta(\mathcal{S} \times \mathcal{A})$. We also denote the conditional probability $\mu(a|s)$ by $\pi_\beta(a|s)$ and call $\pi_\beta(a|s)$ the behavior policy. However, $\mu$ is not assumed to be induced by $\pi_\beta$ for generality. We also use $\mu(s)$ to represent the marginal distribution of state, i.e. $\mu(s) = \sum_{a \in \mathcal{A}} \mu(s, a)$. We also assume access to a dataset $\mathcal{D}_0 = \{s_i\}_{i=1}^{N_0}$ with i.i.d. samples from the initial distribution $\rho$, similar to prior works (Zhan et al., 2022). The goal of offline RL is to learn a policy $\hat{\pi}$ based on the offline dataset so as to minimize the sub-optimality with respect to an optimal policy $\pi^*$, i.e. $J(\pi^*) - J(\hat{\pi})$ with high probability.

**Marginalized importance sampling.** In this paper, we consider primal-dual formulation of LP that aims at learning weights $w(s, a)$ to represent discounted stationary distribution when multiplied by data distribution: $d_w(s, a) = w(s, a)\mu(s, a)$. Also denote $d_w(s) = \sum_{a \in \mathcal{A}} d_w(s, a)$. We define the policy induced by $w$ as $\pi_w(a|s) = d_w(s, a)/d_w(s)$ for $d_w(s) > 0$ and $\pi_w(a|s) = 1/|\mathcal{A}|$ for $d_w(s) = 0$. Typically, $d_w$ is not necessarily equal to $d^{\pi_w}$ for any $w$, but it holds true for the true density ratio of any $\pi$.

## 4 METHOD

This section is divided into three parts. Section 4.1 introduces the primal-dual formulation of offline RL, along with the challenges associated with solving it. We will learn the optimal supported dataset based on this formulation. Section 4.2 presents OSD-DICE algorithm, which generates a near-optimal supported distribution with polynomial sample complexity under general function approximation and single-policy concentrability assumptions. In Section 4.3, we describe how to combine the learned near-optimal distribution and the resulting dataset with representative behavior regularization-based offline RL algorithms, and show how this leads to safe policy improvement beyond the optimal supported policy. The proofs for all of the theoretical results are included in Appendix A.

### 4.1 PRIMAL-DUAL FORMULATION OF LP FOR OFFLINE RL

We start with the following regularized dual formulation of LP for policy optimization (Lee et al., 2021)

$$\max_{d \geq 0} \mathbb{E}_{s,a \sim d}\left[r(s,a)\right] - \alpha \mathbb{E}_{(s,a)\sim\mu}\left[f\left(\frac{d(s,a)}{\mu(s,a)}\right)\right]$$
$$\textbf{s.t.} \quad d(s) = (1-\gamma)\rho(s) + \gamma \sum_{s',a'} P(s|s',a')d(s',a') \ \ \forall s \in \mathcal{S}. \tag{1}$$

where $\mathbb{E}_{(s,a)\sim\mu}\left[f\left(\frac{d(s,a)}{\mu(s,a)}\right)\right] = D_f(d\|\mu)$ is the $f$-divergence between the learning $d$ and the dataset distribution $\mu$, with $f$ being some strictly convex and continuously differentiable function, and $\alpha$ is a hyper-parameter used to control the degree of closeness between the two distributions. Once the optimal stationary distribution is obtained, one can recover the optimal policy from the optimal stationary distribution easily. The constrained problem (1) can be converted into an unconstrained problem by using Lagrangian multiplier $\nu \in \mathbb{R}^S$ and replacing $d$ with the density ratio $w$ (Lee et al., 2021; Zhan et al., 2022).

$$\min_{\nu} \max_{w \geq 0} L_\alpha(w,\nu) = (1-\gamma)\mathbb{E}_{s\sim\rho}[\nu(s)] + \mathbb{E}_{(s,a)\sim\mu}[w(s,a)e_\nu(s,a)] - \alpha\mathbb{E}_{(s,a)\sim\mu}[f(w(s,a))], \tag{2}$$

where $e_\nu(s,a) := r(s,a) + \gamma \sum_{s'} P(s'|s,a)\nu(s') - \nu(s)$ and we refer to it as the advantage. We denote the optimum of (2) as $(\nu_\alpha^*, w_\alpha^*)$. When $\alpha = 0$, $\nu_0^*$ is exactly the optimal state-value function $V^{\pi^*}$, and $d_0^* := w_0^* \cdot \mu$ is the discounted stationary distribution of the optimal policy.

We can show that for any $w$, the associated stationary distribution $d^{\pi_w}$ of $\pi_w$ (defined in Section 3) is supported by the data distribution $\mu$, with the proof given in Appendix A.1.

**Proposition 1** (in-support property). *For any bounded $w$, $d^{\pi_w}(s,a) = 0$, for all the $(s,a)$ pairs satisfying $\mu(s,a) = 0$.*

Our goal is to learn the optimal $w_\alpha^*$, which satisfies $d_{w_\alpha^*} = d^{\pi_{w_\alpha^*}}$ exactly as $w_\alpha^*$ is the true density ratio induced by the natural policy $\pi_{w^*}$, and thus enables us to apply importance weights $w_\alpha^*$ to reweigh the distribution $\mu$. This is the primary reason for adopting this formulation since it facilitates the natural generation of an in-support near-optimal stationary distribution. Once we reweigh $\mathcal{D}$ by $w_\alpha^*$, we can obtain a sampled dataset from $d_{w_\alpha^*}$, which is referred to as the optimal supported dataset, denoted by $\mathcal{D}_{\text{osd}}$. In the following text, for the sake of simplicity, we will use $d_\alpha^*$ to represent $d_{w_\alpha^*}$.

To avoid the numerical instability and poor convergence issue (Goodfellow et al., 2020) caused by directly optimizing the minimax optimization (2), Lee et al. (2021) proposes to substitute the closed-form solution of the inner optimization $w_\nu^*(s,a)$ into (2), with

$$w_\nu^*(s,a) = \max\left(0, (f')^{-1}\left(\frac{e_\nu(s,a)}{\alpha}\right)\right), \tag{3}$$

where $(f')^{-1}$ is the inverse function of the derivative $f'$ of $f$, so that the overall problem is reduced into a single optimization problem:

$$\min_{\nu} L_\alpha\left(w_\nu^*,\nu\right) = (1-\gamma)\mathbb{E}_{s\sim\rho(s)}[\nu(s)] + \mathbb{E}_{(s,a)\sim\mu}\left[-\alpha f\left(\max\left(0, (f')^{-1}\left(\frac{1}{\alpha}e_\nu(s,a)\right)\right)\right)\right]$$
$$+ \mathbb{E}_{(s,a)\sim\mu}\left[\max\left(0, (f')^{-1}\left(\frac{1}{\alpha}e_\nu(s,a)\right)\right)(e_\nu(s,a))\right] \tag{4}$$

Lee et al. (2021) optimizes the empirical version of $L_\alpha(w_\nu^*, \nu)$, denoted as $\widehat{L}_\alpha(w_\nu^*, \nu)$, and utilizes a single-transition estimation $\tilde{e}_\nu = r(s, a) + \gamma\nu(s') - \nu(s)$ to approximate $e_\nu$. However, due to the non-linearity and double-sample problem present in the objective, this gives rise to a biased estimate of $L_\alpha(w_\nu^*, \nu)$ for stochastic MDPs, as discussed in Lee et al. (2021). As a result, whether the minimum of $\widehat{L}_\alpha(w_\nu^*, \nu)$ is the true optimum of $L_\alpha(w_\nu^*, \nu)$ remains unknown. In the following section, we will present a practical algorithm that tackles the aforementioned issues and helps obtain a near-optimal $\widehat{w}_\alpha$ for $L_\alpha(w_\nu^*, \nu)$,

## 4.2 OSD-DICE Method

Instead of solving (4) directly, we propose to solve

$$\min_{\nu \in \mathcal{V}} L_\alpha(\nu) := L_\alpha(w_\nu^*, \nu) + \mathbb{E}_{(s,a)\sim\mu}\mathbf{1}_{w_\nu^*(s,a)>0}[e_\nu(s,a)^2], \tag{5}$$

where an additional squared regularization term is introduced, and $\mathbf{1}_A$ denotes a indicator function satisfying $\mathbf{1}_A = 1$ if $A$ is true else 0. The regularization term, as proven in Lemma 4, does not alter the optimal solution of (4). However, it does introduce certain properties resembling strong convexity and plays a crucial role in guaranteeing that the solution obtained from optimizing $L_\alpha(\nu)$ is an approximate global optimum. Conversely, if we were to eliminate the regularization term, the resulting optimization objective $L_\alpha(w_\nu^*, \nu)$ would lack strong convexity, thus he behavior of the objective function near the optimal value may be too flat, making it difficult to bound the difference between the learned solution and the optimal solution, potentially leading to an undesired solution.

We optimize the empirical version of $L_\alpha(\nu)$:

$$\min_{\nu \in \mathcal{V}} \widehat{L}_\alpha(\nu) := \widehat{L}_\alpha(w_\nu^*, \nu) + \frac{1}{N}\sum_{j=1}^{N}[\widehat{e}_\nu(s_j, a_j)]^2 \cdot \mathbf{1}_{\widehat{w}_\alpha(s_j, a_j)>0}, \tag{6}$$

and $\widehat{L}_\alpha(w_\nu^*, \nu)$ is the empirical version of $L_\alpha(w_\nu^*, \nu)$ with

$$\widehat{L}_\alpha(w_\nu^*, \nu) = (1-\gamma)\frac{1}{N_0}\sum_{j=1}^{N_0}[\nu(s_{0,j})] + \frac{1}{N}\sum_{j=1}^{N}\left[-\alpha f\left(\max\left(0, (f')^{-1}\left(\frac{1}{\alpha}\widehat{e}_\nu(s_j, a_j)\right)\right)\right)\right]$$

$$+ \frac{1}{N}\sum_{j=1}^{N}\left[\max\left(0, (f')^{-1}\left(\frac{1}{\alpha}\widehat{e}_\nu(s_j, a_j)\right)\right)\widehat{e}_\nu(s_j, a_j)\right]. \tag{7}$$

Here, $\widehat{w}_\alpha(s, a) = \max\left(0, (f')^{-1}\left(\frac{\widehat{e}_\nu(s,a)}{\alpha}\right)\right)$ and we let $\widehat{e}_\nu(s, a) := r(s, a) + \gamma\sum_{s'}\widehat{P}(s'|s, a)\nu(s') - \nu(s)$, with $\widehat{P}$ being the maximum likelihood estimate of the transition model learned from a function class $\mathcal{P}$ using an additional dataset $\mathcal{D}_m$. This treatment effectively mitigates the issue of bias mentioned earlier by ensuring that the empirical objective and the expected objective can be closely controlled, particularly when the sample size $N$ is sufficiently large. Moreover, the error associated with this treatment is solely determined on of $|\mathcal{P}|$ and $N$. The overall algorithm process of OSD-DICE is put in Algorithm 1.

---

**Algorithm 1** OSD-DICE

---

**Inputs:** A function $f$, datasets $\mathcal{D}, \mathcal{D}_0, D_m$, functions classes $\mathcal{V}, \mathcal{P}$.
Estimate transitions via maximum likelihood: $\widehat{P} = \arg\max_{P\in\mathcal{P}}\sum_{i=1}^{N_m}\log P(s_i' \mid s_i, a_i)$
Find a solution $\widehat{\nu}_\alpha$ to (6)
Compute $\widehat{w}_\alpha = \max\left(0, (f')^{-1}\left(\frac{\widehat{e}_{\widehat{\nu}_\alpha}}{\alpha}\right)\right)$
**Return:** $\widehat{w}_\alpha, \widehat{\nu}_\alpha$.

---

### 4.2.1 Theoretical Results for OSD-DICE

We will provide the sample complexity of OSD-DICE under the following assumptions. All involved constants are parameterized by $\alpha$ following the same treatment of (Zhan et al., 2022).

**Assumption 1.** *(Realizability of $\mathcal{V}$, $\mathcal{P}$) Suppose $\nu_\alpha^* \in \mathcal{V}$, $P^* \in \mathcal{P}$, where $\mathcal{V}$ and $\mathcal{P}$ are function classes.*

**Assumption 2.** *(Boundedness of $\mathcal{V}$) Suppose $\nu(s) \leq B_{\nu,\alpha}$, for any $s \in \mathcal{S}, \nu \in \mathcal{V}$.*

**Assumption 3.** *(Properties of $f$) Suppose $f$ satisfies the following properties:*
*(i): $f$ is strongly convex.*
*(ii): $(f')^{-1}(x)$ is $B_{f',\alpha}$-continuous for $|x| \in [0, B_{e,\alpha}/\alpha]$, where $B_{e,\alpha} := (1+\gamma)B_{\nu,\alpha} + 1$. Denote the bound of $|(f')^{-1}(x)|$ as $B_{w,\alpha}$ in this domain.*
*(iii): $0 < f(x) \leq B_{f,\alpha}$ on $0 \leq x \leq B_{w,\alpha}$.*

**Remark 1.** *Since the input of $(f')^{-1}(x)$ is $\widehat{e}_\nu/\alpha$ in $\widehat{L}_\alpha(\nu)$, we specify the domain of $(f')^{-1}(x)$ in (ii) through bounding $\widehat{e}_\nu$. Then the domain of $f(x)$ in $\widehat{L}_\alpha(\nu)$ is $[0, B_{w,\alpha}]$, as shown in (iii). It is also straightforward to verify that the commonly used $\chi^2$-divergence with $f(x) = \frac{1}{2}(x-1)^2$ satisfies Assumption 3.*

**Assumption 4.** *($\pi_\alpha^*$-concentrability ) $\frac{d_\alpha^*(s,a)}{\mu(s,a)} \leq B_{w,\alpha}, \forall s \in \mathcal{S}, a \in \mathcal{A}$.*

**Remark 2.** *It is much weaker than all-policy concentrability (Munos & Szepesvári, 2008; Chen & Jiang, 2019). Here we reuse $B_{w,\alpha}$ from (ii) of Assumption 3, since $d_\alpha^*/\mu = w_\alpha^* = \max\left(0, (f')^{-1}\left(\frac{e_{\nu_\alpha^*}(s,a)}{\alpha}\right)\right) \leq B_{w,\alpha}$ as assumed in (ii).*

**Assumption 5.** *(Continuity for $\widehat{L}_\alpha(\nu)$) Given some $\delta \in (0,1)$, for any $\nu \in \mathcal{V}$, $\mu(s,a)e_\nu(s,a) \notin \mathcal{B}(\alpha f'(0), \epsilon_m)$, where $\epsilon_m = 2\sqrt{\frac{\log 3|\mathcal{P}|/\delta}{N_m}}$ and $\mathcal{B}(x,r)$ is a ball with $x$ as the center and $r$ as the radius.*

**Remark 3.** *Assumption 5 is a technical assumption to ensure the continuity of $\widehat{L}_\alpha(\nu)$ for all $\nu \in \mathcal{V}$. We remark that $\nu_\alpha^*$ satisfies the conditions in Assumption 5 given that $e_{\nu_\alpha^*}(s,a) \neq \alpha f'(0)$, $\forall(s,a) \in \{(s,a) : w_\alpha^*(s,a) = 0\}$ and $N_m$ is large enough. At this time, a neighborhood of $\nu^*$ also belongs to $\mathcal{V}$. Besides, the constraints on $\mathcal{V}$ gradually decrease as $N_m$ increases.*

With above assumptions and notation, we present sample complexity of OSD-DICE.

**Theorem 4** (Sample Complexity of OSD-DICE). *Fix some $\alpha > 0$. Suppose Assumptions 1-5 hold for the said $\alpha$. Then with at least probability $1 - \delta$, the output of Algorithm 1 satisfies*
$$\|d_{\widehat{w}_\alpha} - d_\alpha^*\|_1 \leq \|\widehat{w}_\alpha - w_\alpha^*\|_{2,\mu} \leq \mathcal{E}_{N,N_0,N_m,\alpha}, \tag{8}$$
*where $\mathcal{E}_{N,N_0,N_m,\alpha}$ is a polynomial about $N, N_0, N_m, \alpha$ and its specific form is defined in (54) in Appendix A.*

**Remark 5.** *Theorem 4 shows that Algorithm 1 can obtain a near-optimal distribution for regularized problem (2) with polynomial sample complexity. For a simple choice of $f(x) = \frac{(x-1)^2}{2}$, assume that $B_{\nu,\alpha} \geq 1$, $\alpha \leq 1$ and $N_0 = N_m = N$, then $\mathcal{E}_{N,N_0,N_m,\alpha} = \tilde{O}\left(\frac{B_{\nu,\alpha}^{3/2}}{\alpha^{3/2}} N^{-1/4}\right)$, which leads to the sample complexity $\tilde{O}\left(\frac{B_{\nu,\alpha}^6}{\alpha^6 \epsilon^4}\right)$.*

Furthermore, OSD-DICE is capable of efficiently learning a near-optimal distribution even for the unregularized problem (i.e. $\alpha = 0$) by carefully controlling the magnitude of $\alpha$. The sample complexity of OSD-DICE for the unregularized setting is well-characterized by the following theorem:

**Theorem 6** (Sample complexity of competing with $d_0^*$). *For any $\epsilon > 0$. Suppose that $r := \min r(s,a) > 0$, and there exists $d_0^*$ that satisfies Assumption 4 with $\alpha = 0$. Besides, assume that Assumption 1-5 hold for $\alpha = \alpha_\epsilon := \frac{\epsilon}{2r \cdot B_{f,0}}$ and $f(x) = \frac{(x-1)^2}{2}$. Then if $N = N_m = N_0 = \tilde{O}\left(\frac{B_{\nu,\alpha_\epsilon}^6}{\epsilon^6}\right)$, the output of Algorithm 1 with input $\alpha = \alpha_\epsilon$ satisfies*
$$\|d_0^* - \widehat{d}_{\alpha_\epsilon}\|_1 \leq \epsilon, \tag{9}$$
*with at least probability $1 - \delta$.*

### 4.3 BEHAVIOR REGULARIZATION-BASED OFFLINE RL WITH OPTIMAL SUPPORTED DATASET

With the help of the resulting high-quality distribution $d_{\widehat{w}_\alpha}$ obtained in Section 4.2, conventional offline RL methods based on behavior regularization can be further enhanced. In this section, we instantiate this idea with two representative algorithms and show safe policy improvement over the optimal $\pi_\alpha^*$. In fact, safe policy improvement over $\pi_0^*$ can also be established in a a similar manner, provided that $\alpha$ is small enough, we omit it for the sake of simplicity.

### 4.3.1 BEHAVIOR CLONING WITH OPTIMAL SUPPORTED DATASET

One straightforward behavior regularization method is Behavior Cloning(BC). We can perform BC on $d_{\widehat{w}_\alpha}$, which also serves as the policy extraction phase as follows.

$$\pi_{\widehat{w}_\alpha}(a|s) = \begin{cases} \widehat{w}_\alpha(s,a)\mu(s,a)/\sum_{\bar{a}}\widehat{w}_\alpha(s,\bar{a})\mu(s,\bar{a}), & \text{for } \sum_{\bar{a}}\widehat{w}_\alpha(s,\bar{a})\mu(s,\bar{a}) > 0, \\ 1/|\mathcal{A}|, & \text{else} \end{cases} \tag{10}$$

In practice, given the policy space $\Pi \subseteq (\mathcal{S} \to \Delta(\mathcal{A}))$, we optimize the log-likelihood on $\mathcal{D}_{\text{osd}}$ to obtain $\widehat{\pi}_\alpha^{\text{osd-BC}}$ by

$$\widehat{\pi}_\alpha^{\text{osd-BC}} = \arg\max_{\tilde{\pi}\in\Pi} \sum_{i=1}^{N} \widehat{w}_\alpha(s_i, a_i) \log \tilde{\pi}(a_i|s_i), \tag{osd-BC}$$

**Theorem 7** (Sample complexity of osd-BC). *Fix $\alpha > 0$. Suppose Assumption 1-5 hold for the said $\alpha$. Besides, we further assume that for all $\nu \in \mathcal{V}$ and their induced approximate optimal $\widehat{w}_\nu$, $\pi_{\widehat{w}_\nu} \in \Pi$. Let $\mathcal{E}_N' = \sqrt{\frac{\log|\Pi||\mathcal{V}|/\delta}{N}}$. Then with at least probability $1 - 2\delta$,*

$$J(\pi_\alpha^*) - J(\widehat{\pi}_\alpha^{\text{osd-BC}}) \leq \frac{1}{1-\gamma}\mathbb{E}_{s\sim d_\alpha^*}\left[\left\|\pi_\alpha^*(\cdot\mid s) - \widehat{\pi}_\alpha^{\text{osd-BC}}(\cdot\mid s)\right\|_1\right] \leq \frac{3}{1-\gamma}\mathcal{E}_{N,N_0,N_m,\alpha} + \frac{2}{(1-\gamma)^2}\mathcal{E}_N'. \tag{11}$$

**Remark 8.** *Similar to Remark 5, the sample complexity of osd-BC under quadratic $f$ is $\tilde{O}\left(\frac{B_{\nu,\alpha}^6}{\alpha^6(1-\gamma)^4\epsilon^4}\right)$. Besides, under the same conditions in Theorem 6, the sample complexity of competing with $\pi_0^*$ is $\tilde{O}\left(\frac{B_{\nu,\alpha_\epsilon}^6}{(1-\gamma)^4\epsilon^6}\right)$.*

### 4.3.2 BEHAVIOR REGULARIZATION WITH OPTIMAL SUPPORTED DATASET

While BC-based methods are capable of finding the near-optimal policy supported by $\mu$, they may overlook policies with higher performance that not yet covered by $\mu$. In fact, a class of generalized behavior regularization-based methods as proposed in (Kumar et al., 2020; Cheng et al.), can optimize policies across a much wider space and potentially discover superior policies, if the Bellman error is small on $\mu$ and generalizes well beyond the support of $\mu$. Hence, we propose to adopt one representative formulation of this class: minimax offline RL with relative pessimism (Cheng et al.) on $d_{\widehat{w}_\alpha}$ and give rise to osd-MiniMax formulation.

Concretely, we still search for good policies from a policy class $\Pi$ but additionally assume access to a value function class $\mathcal{F} \subseteq (\mathcal{S} \times \mathcal{A} \to [0, 1/(1-\gamma)])$ to model the Q-functions of policies. osd-MiniMax solves the following bi-level optimization problem. Intuitively, $\widehat{\pi}_\alpha^{\text{osd-minimax}}$ attempts to maximize the value predicted by $f^\pi$, and $f^\pi$ performs a relatively pessimistic policy evaluation of a candidate $\pi$ with respect to the near-optimal supported policy $\pi_{\widehat{w}_\alpha}$.

$$\widehat{\pi}_\alpha^{\text{osd-MiniMax}} = \underset{\pi\in\Pi}{\text{argmax}}\,\mathbb{E}_{d_{\widehat{w}_\alpha}}[f^\pi(s,\pi) - f^\pi(s,a)] \tag{12}$$

$$\text{s.t. } f^\pi = \underset{f\in\mathcal{F}}{\text{argmin}}\,\mathbb{E}_{d_{\widehat{w}_\alpha}}[f(s,\pi) - f(s,a)] + \beta\mathbb{E}_{d_{\widehat{w}_\alpha}}[((f - \mathcal{T}^\pi f)(s,a))^2], \tag{13}$$

where $\beta$ is a hyperparameter, and $f(s,\pi) = \sum_a \pi(a|s)f(s,a)$. The following theorem shows that $\widehat{\pi}_\alpha^{\text{osd-MiniMax}}$ is a safe policy improvement over the optimal policy $\pi_\alpha^*$.

**Theorem 9** (Safe improvement over $\pi_\alpha^*$ for osd-MiniMax). *Suppose that Assumptions 1-5 hold, and for all $\pi \in \Pi$, $Q^\pi \in \mathcal{F}$, and $\pi_{\widehat{w}_\alpha} \in \Pi$, then with at least probability $1 - \delta$, we have $J(\widehat{\pi}_\alpha^{\text{osd-MiniMax}}) \geq J(\pi_\alpha^*) - \frac{2}{1-\gamma}\mathcal{E}_{N,N_0,N_m,\alpha}$.*

In practice, osd-MiniMax can be achieved by implementing CQL (Kumar et al., 2020) on $\mathcal{D}_{\text{osd}}$, named osd-CQL. The conservative value evaluation in osd-CQL can be modeled by (13) with $1/\beta$ the conservative coefficient, and the policy improvement in osd-CQL can be modeled by (12). Although osd-CQL maximizes $f^\pi(s,\pi)$ instead of $f^\pi(s,\pi) - f^\pi(s,a)$ for policy improvement, the two are essentially equivalent as the minus term in equation (12) is irrelevant to $\pi$.

**Theorem 10** (Safe improvement over $\pi_\alpha^*$ for osd-CQL, informal). *Suppose that Assumptions 1-5 hold. Let $\widehat{\pi}^{osd\text{-}CQL}$ be the policy obtained by osd-CQL. Then with at least $1 - 2\delta$, $\widehat{\pi}^{osd\text{-}CQL}$ satisfies*

$$J(\widehat{\pi}^{osd\text{-}CQL}) \geq J(\pi_\alpha^*) - \frac{2}{1-\gamma}\mathcal{E}_{N,N_0,N_m,\alpha} - sampling\ error + \alpha_{cql} \cdot positive\ term \qquad (14)$$

*where $\alpha_{cql}$ is the conservative coefficient.*

## 5 EXPERIMENTAL EVALUATION

### 5.1 RANDOM MDPS (TABULAR MDPS)

We begin by evaluating OSD-DICE on randomly generated MDPs. We employ randomly generated MDPs with a state space size $|\mathcal{S}| = 60$, an action space size $|\mathcal{A}| = 8$, a discount factor of $\gamma = 0.95$. We then formulate a data-collection policy aligned with a predefined degree of optimality, $\zeta \in \{0.8, 0.9\}$. Following this, we collect $N$ trajectories (where $N \in \{200, 300, 400, 500, 600, 700, 800\}$) from the generated MDP and the associated data-collection policy $\pi_\mu$. Subsequently, the accumulated trajectories are given to offline RL algorithm. The candidate transition model class $\mathcal{P}$ is set to contain the true transition model $P^*$. We then evaluate the algorithm's performance, both in terms of the mean performance and the Conditional Value at Risk (CVaR) at the 5% threshold, which considers the worst 5% runs. Experimental details are put in Appendix B.

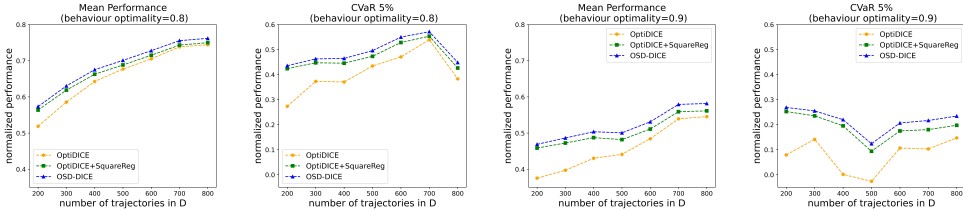

Figure 1: Comparison between OSD-DICE, OptiDICE and OptiDICE with squared regularization in random MDPs

We compared OSD-DICE with OptiDICE (Lee et al., 2021) and "OptiDICE+SquareReg" in Figure 1 to examine the individual contributions of the approximated advantage estimator and squared regularization. The comparison between the yellow and green curves demonstrates that using squared regularization alone outperforms OptiDICE in terms of both mean performance and CVaR, especially in scenarios with higher levels of behavior optimality. This indicates a significantly positive impact of squared regularization. Furthermore, comparing the green and blue curves reveals that incorporating the approximated advantage estimator further enhances the mean performance and CVaR. This confirms the distinct benefits of using an approximated advantage estimator.

### 5.2 D4RL BENCHMARK

We evaluate OSD-DICE on continuous MDPs using the D4RL offline RL benchmarks (Fu et al., 2020), Maze2D (3 datasets), Gym-MuJoCo (12 datasets) and Antmaze (4 datasets) are selected from the D4RL dataset. Please note that in this setting, we have chosen to use the single-transition estimator to approximate $e_\nu$. This decision is based on the fact that the MDPs involved in these tasks are approximately deterministic, and it is discussed in Lee et al. (2021) that the bias issue is not a concern in this situation. Therefore, we can rely on the single-transition estimator to provide reliable results in this situation. Implementation details, experimental configurations and hyperparameter selection can be found in Appendix C.

**Comparison against OptiDICE.** We compare OSD-BC with OptiDICE in Appendix E Table 3. We also compare OSD-CQL with OptiDICE-CQL, where density ratio weights are obtained from OptiDICE, in Appendix D Figure 2 (b) ($\beta = 0$.). The results indicate that in Maze2d domain, OSD-BC performs similarly to OptiDICE. In Locomotion domain, OSD-BC significantly outperforms OptiDICE. When combined with CQL, OSD-CQL outperforms OptiDICE-CQL in Maze2d, Locomotion, and Antmaze domain. This confirms the role of the regularization term.

**Comparison against other Reweighing Methods.** On the D4RL benchmarks, we implement OSD method on top of BC and two representative behavior regularization-based algorithms TD3BC(Fujimoto & Gu, 2021) and CQL (Kumar et al., 2020). We compare OSD against different reweighing strategies in Table 1, including uniform strategy, which is the original algorithm, "top%10" method, which picks the top10% of trajectories based on cumulative reward to learn, AW(Hong et al.) and RW (Hong et al.), which leverage exponential advantage and exponential return to reweigh. We also compare osd-BC with some other weighted-BC approaches, see Table 3 in Appendix E, and compare osd-CQL with some other state-of-the-art reweighing approaches, see Table 4 in Appendix F.

Table 1: Averaged normalized scores on MuJoCo locomotion, Maze2d and Antmaze tasks over three seeds. Note that unif=uniform, M=Maze2d, Ho=Hopper, Ha=Halfcheetah, W=Walker2d, A=Antmaze.

| | BC | | | | | TD3BC | | | | | CQL | | | | |
|---|---|---|---|---|---|---|---|---|---|---|---|---|---|---|---|
| | unif | 10% | AW | RW | OSD(ours) | unif | 10% | AW | RW | OSD(ours) | unif | 10% | AW | RW | OSD(ours) |
| M-u | 5.5 | -13.2 | 6.8 | 5.2 | **123.3±23** | 31.4 | -2.8 | 62.0 | 66.6 | **70.±24** | -14.5 | 12.5 | 20.7 | 13.5 | **148.4±33.** |
| M-m | 11.4 | 16.5 | 14.9 | 12.7 | **81.0±9.4** | 26.3 | 51.0 | 43.2 | 47.8 | **105.8±34** | 27.8 | 5.5 | -2.9 | 32.0 | **102.5±13.0** |
| M-l | -0.4 | 6.1 | 10.3 | 8.9 | **154.0±21.4** | 130.1 | 71.2 | 68.7 | 52.7 | **164.2±42** | -1.8 | 14.2 | 9.9 | 1.7 | **132.2±13.3** |
| M-total | 16.6 | 9.4 | 32.1 | 26.8 | 358.3 | 189.2 | 119.4 | 173.9 | 189.2 | 340.9 | 11.5 | 32.2 | 27.7 | 47.2 | 383.1 |
| Ho-r | 3.9 | 4.3 | 4.6 | 4.6 | **31.5±0.3** | 8.4 | 7.8 | 4.5 | 4.1 | **13.2** | 9.2 | 10.1 | 7.2 | 2.3 | **32.6±0.4** |
| Ho-m | 54.0 | 56.3 | 57.1 | 56.5 | **59.7±3.1** | 59.7 | 65.2 | 62.7 | 61.2 | **69.2±4.3** | 59.1 | 64.2 | 68.5 | 63.9 | **88.6±1.9** |
| Ho-m-r | 27.9 | 70.4 | 69.2 | 71.4 | 35.8±3.9 | 64.1 | 91.2 | 93.2 | 90.0 | **93.2±13.1** | 95.6 | 93.2 | 93.6 | 93.7 | **100.8±1.0** |
| Ho-m-e | 51.2 | 110.0 | 110.7 | 109.3 | 95.3±8.0 | 95.4 | 107.7 | 111.0 | 110.5 | **110.9±2.9** | 104.5 | 105.5 | **110.4** | 110.0 | 109.0±4.7 |
| Ha-r | 2.3 | 1.8 | 2.2 | 2.0 | **5.1±1.3** | 12.2 | 10.3 | 11.2 | 13.1 | **16.7±0.1** | 21.3 | 2.4 | 9.5 | 7.8 | **27.5±0.1** |
| Ha-m | 42.2 | 42.2 | 42.0 | 42.3 | **42.5±0.4** | 48.3 | 45.3 | 48.6 | 48.3 | **51.2±0.5** | 48.5 | 45.2 | 48.2 | 49.0 | **59.5±0.45** |
| Ha-m-r | 35.8 | 26.1 | 39.6 | 39.2 | 39.5±2.4 | 44.6 | 42.2 | 45.0 | 45.5 | **45.9±0.7** | 47.0 | 42.5 | 44.6 | 45.3 | **51.5±0.2** |
| Ha-m-e | 56.3 | 92.1 | 92.0 | 91.5 | 85.8±3.8 | 88.8 | 73.4 | **97.7** | 97.6 | 89.2±4.8 | 93.0 | 78.2 | 87.5 | 80.0 | **93.7±4.1** |
| W-r | 1.4 | 1.3 | 1.3 | 1.1 | **5.8±2.8** | 1.1 | **2.3** | 1.0 | 1.6 | 2.2±0.1 | 5.1 | **9.3** | 4.7 | 0.2 | 5.6±1.2 |
| W-m | 65.7 | 70.2 | 70.1 | 64.9 | **73.0±4.3** | 84.3 | 78.2 | 82.2 | 81.5 | **82.0±2.8** | 82.2 | 76.7 | 82.3 | 75.4 | **83.3±1.0** |
| W-m-r | 16.7 | 54.5 | 55.3 | 47.5 | **56.2±5.8** | 80.1 | 71.8 | 77.8 | 69.9 | **89.5±2.9** | 71.3 | 75.0 | 78.1 | 62.0 | **86.3±8.2** |
| W-m-e | 96.5 | **108.4** | 107.1 | 107 | 107.4±2.1 | 109.9 | 108.9 | 110.1 | 110.2 | **110.5±0.1** | 108.9 | 108.9 | 109.3 | 108.6 | **110.5±0.1** |
| L-total | 454.4 | 633.9 | 657.3 | **665.5** | 637.6 | 696.8 | 699.5 | 745.0 | 732.6 | **773.8** | 745.7 | 711.2 | 743.9 | 698.2 | **848.9** |
| A-u | 53.5 | 60.0 | 54.1 | 61.0 | **85.0±14.5** | 17.3 | 53.6 | 43.0 | 46.6 | **66.7±15.0** | 76.0 | 69.3 | 77.0 | 72.0 | **91.0±7** |
| A-u-d | 45.0 | 43.5 | 44.0 | 50.0 | **65.0±6.5** | 64.6 | 29.0 | 67.3 | 63.3 | **70.7±8.3** | 48.0 | 24.7 | 36.0 | 43.0 | **55.7±5.5** |
| A-m-d | 0.0 | 36.0 | 22. | 28.0 | 10.0±2.5 | 3.6 | 3.3 | 0.0 | **3.6** | 0.3±0.5 | 0.0 | 0.0 | 6.0 | 23.0 | **42.0±6.9** |
| A-m-p | 0.0 | 43.3 | 22. | 22. | 12.0±{5.5} | 0.0 | 2.0 | 0.3 | 0.0 | 1.0±0 | 2.4 | 0.0 | 10.7 | 26.0 | **39.0±8.5** |
| A-total | 98.5 | 182.8 | 142.0 | 161.0 | 172.0 | 85.5 | 87.9 | 110.6 | 110.9 | 138.7 | 126.4 | 94.0 | 129.7 | 164.0 | 227.7 |

We can draw the following conclusions from Table 1. Firstly, OSD-embedded algorithms outperform their corresponding original algorithms in all three classes of tasks. This indicates that OSD effectively distinguishes the importance level of samples, and the usage of a near-optimal supported dataset significantly boosts the performance of the original algorithms. Next, when OSD is combined with BC, OSD-BC performs significantly better than other reweighing baselines on Maze2d task and performs comparably to these baselines on Locomotion and Antmaze tasks. When combined with TD3BC and CQL, OSD algorithm outperforms other reweighing methods in all three classes of tasks. In particular, the advantage is more pronounced when combined with CQL, which is consistent with the theoretical results. Lastly, while top10% strategy is a competitive method when combined with BC, its effectiveness does not improve when combined with behavior regularization-based methods. In fact, it even shows a significant decline in performance on some tasks. This suggests that although it can improve data quality, continuing with policy optimization becomes difficult due to the reduction in the total amount of data.

## 6 CONCLUSION AND FUTURE WORK

We propose to perform behavior regularization-based offline RL on an optimal supported dataset rather than the original dataset. This approach encourages the learned policy to stay within the support of the dataset while minimizing the potential bias towards the optimization objective. To obtain the optimal supported dataset, we introduce OSD-DICE, which has a proven polynomial sample complexity under general function approximation and single-policy concentrability. We also show safe policy improvement over the near-optimal supported policy when using representative behavior regularization-based offline RL methods. Our empirical results demonstrate the effectiveness of OSD-DICE on tabular tasks and the performance enhancement it induces on D4RL benchmarks. In our future work, we aim to extend the application of our proposed framework to a broader range of offline RL methods and more complex tasks, while further refining the sample complexity.

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

# A    PROOFS

## A.1    PROOF OF PROPOSITION 1

To proceed, we divide the state-action space $\mathcal{Z} := \mathcal{S} \times \mathcal{A}$ into $\mathcal{Z}_{in} := \{(s,a)|\mu(s,a) > 0\}$ and $\mathcal{Z}_{out} := \{(s,a), \mu(s,a) = 0\}$.

**Lemma 1.** *If $(s,a) \in \mathcal{Z}_{out}$, then $\rho(s)\pi_w(a|s) = 0$ for any $w \in W$.*

*Proof.* We argue that for any $(s,a) \in \mathcal{Z}_{out}$, either (i) $\mu(s) = 0$ or (ii) $\mu(s) \neq 0$ and $\pi_\beta(a|s) = 0$, where $\pi_\beta$ is the behavior policy defined in Section 3. For case (ii), one has $\pi_w(a|s) = 0$ due to the construction of $\pi_w$, then $\rho(s)\pi_w(a|s) = 0$ holds true straightforwardly. For case (i), we can prove that $\rho(s) = 0$ and then $\rho(s)\pi_w(a|s) = 0$ holds true naturally. We derive $\rho(s) = 0$ by contradiction: if it was wrong, one would had $d^\star(s) \geq (1-\gamma)\rho(s) > 0$, then there existed some $a \in \mathcal{A}$ such that $d^\star(s,a) > 0$, which induced $\frac{d^\star(s,a)}{\mu(s,a)} = \infty$, and this contradicts with Assumption 4.  $\square$

*Proof of Proposition 1.* Since $\mu$ satisfies

$$\mu(s,a) = \sum_{s',a'} \underbrace{\pi_\beta(a|s)P(s|s',a')}_{P^{\pi_\beta}((s',a')\rightarrow(s,a))} \mu(s',a'), \forall(s,a) \in \mathcal{Z}, \tag{15}$$

then for $(s,a) \in \mathcal{Z}_{out}$, one has

$$P^{\pi_\beta}((s',a') \rightarrow (s,a)) = 0, \forall(s',a') \in \mathcal{Z}_{in}. \tag{16}$$

This implies that $\pi_\beta(a|s) = 0$ when $P(s|s',a') > 0$. By the construction of $\pi_w$, we can derive that $\pi_w(a|s) = 0$ when $P(s|s',a') > 0$, which implies that

$$P^{\pi_w}((s',a') \rightarrow (s,a)) = 0, \;\; \forall(s,a) \in \mathcal{Z}_{out}, \forall(s',a') \in \mathcal{Z}_{in}. \tag{17}$$

Combining the conclusion that $\rho(s)\pi_w(a|s) = 0, \forall(s,a) \in \mathcal{Z}_{out}$ from Lemma 1, it can be deduced that $\pi_w$ can visit $\mathcal{Z}_{out}$ neither by starting from the initial distribution $\rho(s)$ nor by starting from $\mathcal{Z}_{in}$, then $d^{\pi_w}(s,a) = 0, \forall(s,a) \in \mathcal{Z}_{out}$.  $\square$

## A.2    PROOFS OF THEOREM 4

Before proceeding, we first present some lemmas to bound the difference between the true transition model and the learned transition model.

**Lemma 2.** *(Convergence of MLE for learning transitions (Geer, 2000)) Given a realizable model class $\mathcal{P}$ that contains the true model $P^*$ and a dataset $\mathcal{D} = \{(s_i, a_i, s_i')\}$ with $(s_i, a_i) \sim \mu, s_i' \sim P^*(\cdot|s_i, a_i)$, let $\widehat{P}$ be*

$$\widehat{P} = \arg\max_{P \in \mathcal{P}} \sum_{i=1}^{N} \log P(s_i'|s_i, a_i). \tag{18}$$

*Fix the failure probability $\delta > 0$. Then, with probability at least $1 - \delta$, we have the following concentration on the squared Hellinger distance between $\widehat{P}$ and $P^*$:*

$$\mathbb{E}_{s,a\sim\mu}\left[\sum_{s'}\left(\sqrt{\widehat{P}(s' \mid s,a)} - \sqrt{P(s' \mid s,a)}\right)^2\right] \lesssim \frac{\log(|\mathcal{P}|/\delta)}{N} \tag{19}$$

**Lemma 3.** *With probability at least $1 - \frac{\delta}{3}$,*

$$\mathbb{E}_\mu \sum_{s'} |\widehat{P}(s'|s,a) - P(s'|s,a)| \leq 2\sqrt{\frac{\log 3|\mathcal{P}|/\delta}{N_m}}, \tag{20}$$

*Proof.* Notice that

$$\mathbb{E}_\mu \sum_{s'} |\widehat{P}(s'|s,a) - P(s'|s,a)| = \sum_{s,a,s'} \mu(s,a)|\widehat{P}(s'|s,a) - P(s'|s,a)|$$

$$= \sum_{s,a,s'} \left( \sqrt{\mu(s,a)}(\sqrt{\widehat{P}(s'|s,a)} + \sqrt{P(s'|s,a)}) \right) \cdot \left( \sqrt{\mu(s,a)}\left| \sqrt{\widehat{P}(s'|s,a)} - \sqrt{P(s'|s,a)} \right| \right)$$

$$\leq \left( \sum_{s,a,s'} \mu(s,a)(\sqrt{\widehat{P}(s'|s,a)} + \sqrt{P(s'|s,a)})^2 \right)^{1/2} \cdot \left( \sum_{s,a,s'} \mu(s,a)(\sqrt{\widehat{P}(s'|s,a)} - \sqrt{P(s'|s,a)})^2 \right)^{1/2}$$

$$\tag{21}$$

$$\leq 2\sqrt{\frac{\log 3|\mathcal{P}|/\delta}{N_m}}, \qquad \text{(with probability at least } 1 - \frac{\delta}{3}) \tag{22}$$

where (21 holds due to Holder inequality, and (22) holds due to Lemma 2. $\qquad \square$

Recall some definitions presented in the main text.

$$e_\nu(s,a) := r(s,a) + \gamma \sum_{s'} P(s'|s,a)\nu(s') - \nu(s) \qquad \text{(expected advantage of } \nu)$$

$$\widehat{e}_\nu(s,a) := r(s,a) + \gamma \sum_{s'} \widehat{P}(s'|s,a)\nu(s') - \nu(s) \qquad \text{(empirical advantage of } \nu)$$

$$w_\nu^*(s,a) := \max\left( 0, (f')^{-1}\left( \frac{1}{\alpha}e_\nu(s,a) \right) \right) \qquad \text{(optimal } w \text{ of } \nu)$$

$$\widehat{w}_\nu(s,a) := \max\left( 0, (f')^{-1}\left( \frac{1}{\alpha}\widehat{e}_\nu(s,a) \right) \right) \qquad \text{(empirical optimal } w \text{ of } \nu),$$

We next give a lemma to show an important property for the optimal $\nu_\alpha^*$ of (2):

**Lemma 4.** *The optimal* $\nu_\alpha^*$, $w_\alpha^*$ *of (2) satisfies*

$$\mathbb{E}_\mu \mathbf{1}_{w_\alpha^*(s,a)>0}[e_{\nu_\alpha^*}(s,a)]^2 = 0.$$

*Proof.* Since $\nu_\alpha^*$ is the optimal solution to the regularized primal-dual program (2), it is equal to the value function (Zhan et al., 2022). Then $e_{\nu_\alpha^*}$ is the optimal advantage function and $\mathbb{E}_{\mu(s,a)}[w_\alpha^*(s,a)e_{\nu_\alpha^*}] = 0$ since it captures the optimal advantage of optimal policy. Therefore, $e_{\nu_\alpha^*} = 0$ on all $(s,a)$ satisfying $w_\alpha^*(s,a) > 0$. So we obtain Lemma 4 straightforwardly. $\qquad \square$

By Lemma 4 we can deduce the following lemma directly:

**Lemma 5.** *The optimal solution to (2) is also the optimal solution to* $L(w_\nu^*, \nu)$.

Next we introduce a new function $h(x)$ by

$$h(x) := -f\left( \max\left( 0, (f')^{-1}(x) \right) \right) + \max\left( 0, (f')^{-1}(x) \right) \cdot x.$$

Define

$$L(w_\nu^*, \nu) = (1-\gamma)\mathbb{E}_{s\sim\rho(s)}[\nu(s)] + \mathbb{E}_{(s,a)\sim\mu}\left[ \alpha h\left( \frac{1}{\alpha}e_\nu(s,a) \right) \right]$$

$$+ \mathbb{E}_\mu \mathbf{1}_{w_\nu^*(s,a)>0}[e_\nu(s,a)]^2. \text{ (expected objective)}$$

$$\widehat{L}_\mathcal{D}(\widehat{w}_\nu, \nu) = (1-\gamma)\frac{1}{N_0}\sum_{j=1}^{N_0}[\nu(s_{0,j})] + \frac{1}{N}\sum_{j=1}^{N}\left[ \alpha h\left( \frac{1}{\alpha}\widehat{e}_\nu(s,a) \right) \right]$$

$$+ \frac{1}{N}\sum_{j=1}^{N}\mathbf{1}_{\widehat{w}_\nu(s,a)>0}[\widehat{e}_\nu(s_j, a_j)]^2, \text{ (empirical objective)}$$

then $\widehat{L}_{\mathcal{D}}\left(\widehat{w}_{\nu}, \nu\right)$ is actually the objective (6) of Algorithm 1.

Throughout the sequel, we will denote the optimal solution to $L\left(w_{\nu}^{*}, \nu\right)$ as $\nu^{*}$ and the optimal solution to $\widehat{L}_{\mathcal{D}}\left(\widehat{w}_{\nu}, \nu\right)$ as $\widehat{\nu}$ for simpilicity, our goal is to bound $w_{\nu^{*}}^{*}$ and $\widehat{w}_{\widehat{\nu}}$.

We further define two intermediate objectives as

$$
L\left(\widehat{w}_{\nu}, \nu\right) = (1-\gamma)\mathbb{E}_{s\sim\rho(s)}[\nu(s)] + \mathbb{E}_{(s,a)\sim\mu}\left[\alpha h\left(\frac{1}{\alpha}\widehat{e}_{\nu}(s,a)\right)\right]
$$
$$
+ E_{\mu}\mathbf{1}_{\widehat{w}_{\nu}(s,a)>0}[\widehat{e}_{\nu}(s,a)]^{2}, \qquad \text{(expected objective with empirical } \widehat{e}_{\nu})
$$

$$
\widehat{L}_{\mathcal{D}}\left(w_{\nu}^{*}, \nu\right) := (1-\gamma)\frac{1}{N_{0}}\sum_{j=1}^{N_{0}}[\nu(s_{0,j})] + \frac{1}{N}\sum_{j=1}^{N}\left[\alpha h\left(\frac{1}{\alpha}e_{\nu}(s_{j},a_{j})\right)\right]
$$
$$
+ \frac{1}{N}\sum_{j=1}^{N}\mathbf{1}_{w_{\nu}^{*}(s_{j},a_{j})>0}[e_{\nu}(s_{j},a_{j})]^{2}. \qquad \text{(empirical objective with expected } e_{\nu})
$$

**Lemma 6.** *Let*

$$
\epsilon_{N,N_{0},N_{m}} := 2\alpha(B_{f,\alpha} + \frac{B_{w,\alpha}B_{e,\alpha}}{\alpha} + B_{e,\alpha}^{2})\sqrt{\frac{2\log 6|\mathcal{V}|/\delta}{N}} + 2B_{\nu,\alpha}\sqrt{\frac{2\log 6\mathcal{V}|/\delta}{N_{0}}}
$$
$$
+ 4(B_{w,\alpha} + 2B_{e,\alpha})B_{\nu,\alpha}\sqrt{\frac{\log 3|\mathcal{P}|/\delta}{N_{m}}}, \tag{23}
$$

*where $B_{e,\alpha} = 1 + (1+\gamma)B_{\nu,\alpha}$. Then*

$$
L\left(w_{\nu}^{*}, \widehat{\nu}\right) - L\left(w_{\nu^{*}}^{*}, \nu^{*}\right) \leq \epsilon_{N,N_{0},N_{m}}.
$$

*Proof.* Decompose the objective difference according to

$$
L\left(w_{\nu}^{*}, \widehat{\nu}\right) - L\left(w_{\nu^{*}}^{*}, \nu^{*}\right) \leq \underbrace{L\left(w_{\nu}^{*}, \widehat{\nu}\right) - L\left(\widehat{w}_{\widehat{\nu}}, \widehat{\nu}\right)}_{T_{1}} + \underbrace{L\left(\widehat{w}_{\widehat{\nu}}, \widehat{\nu}\right) - \widehat{L}_{\mathcal{D}}\left(\widehat{w}_{\widehat{\nu}}, \widehat{\nu}\right)}_{T_{2}}
$$
$$
+ \underbrace{\widehat{L}_{\mathcal{D}}\left(\widehat{w}_{\widehat{\nu}}, \widehat{\nu}\right) - \widehat{L}_{\mathcal{D}}\left(\widehat{w}_{\nu^{*}}, \nu^{*}\right)}_{T_{3}} + \underbrace{\widehat{L}_{\mathcal{D}}\left(\widehat{w}_{\nu^{*}}, \nu^{*}\right) - L\left(\widehat{w}_{\nu^{*}}, \nu^{*}\right)}_{T_{4}}
$$
$$
+ \underbrace{L\left(\widehat{w}_{\nu^{*}}, \nu^{*}\right) - L\left(w_{\nu^{*}}^{*}, \nu^{*}\right)}_{T_{5}}. \tag{24}
$$

It is straightforward that $T_{3} \leq 0$ since $\widehat{\nu}$ is the minimum of $\widehat{L}_{\mathcal{D}}\left(\widehat{w}_{\nu}, \nu\right)$.

• **Upper bounded of $T_{2}$, $T_{4}$**
for any $\nu \in \mathcal{V}$, let $h_{i}^{\nu} = h\left(\frac{1}{\alpha}\widehat{e}_{\nu}(s_{i},a_{i})\right) + \mathbf{1}_{\widehat{w}(s_{i},a_{i})>0}\widehat{e}_{\nu}(s_{i},a_{i})^{2}$. From Assumption 2 we have

$$
|\widehat{e}_{\nu}(s,a)| \leq 1 + (1+\gamma)B_{\nu,\alpha} = B_{e,\alpha}, \tag{25}
$$

then by (ii) and (iii) in Assumption 3 we have

$$
|h_{i}^{\nu}| \leq B_{f,\alpha} + \frac{B_{w,\alpha}B_{e,\alpha}}{\alpha} + B_{e,\alpha}^{2}. \tag{26}
$$

Notice that $h_{i}^{\nu}$ is independent for different $i$, thus using Hoeffding's inequality and and for any $t > 0$,

$$
\Pr\left[|\frac{1}{N}\sum_{i=1}^{N}h_{i}^{\nu} - \mathbb{E}[h_{i}^{\nu}]| \leq t\right] \geq 1 - 2\exp\left(\frac{-Nt^{2}}{2\left(B_{f,\alpha} + \frac{B_{w,\alpha}B_{e,\alpha}}{\alpha} + B_{e,\alpha}^{2}\right)^{2}}\right). \tag{27}
$$

Let $t = \left( B_{f,\alpha} + \frac{B_{w,\alpha}B_{e,\alpha}}{\alpha} + B_{e,\alpha}^2 \right) \sqrt{\frac{2\log\frac{6|\mathcal{V}|}{\delta}}{N}}$, we have with at least probability $1 - \frac{\delta}{3|\mathcal{V}|}$,

$$| \frac{1}{N} \sum_{i=1}^{N} h_i^\nu - \mathbb{E}[h_i^\nu] | \leq \left( B_{f,\alpha} + \frac{B_{w,\alpha}B_{e,\alpha}}{\alpha} + B_{e,\alpha}^2 \right) \sqrt{\frac{2\log\frac{6}{\delta}}{N}}. \tag{28}$$

Therefore by union bound, with at least probability $1 - \frac{\delta}{3}$, we have for all $\nu \in \mathcal{V}$,

$$|\frac{1}{N} \sum_{i=1}^{N} h_i^\nu - \mathbb{E}[h_i^\nu]| \leq (B_{f,\alpha} + \frac{B_{w,\alpha}B_{e,\alpha}}{\alpha} + B_{e,\alpha}^2)\sqrt{\frac{2\log 6|\mathcal{V}|/\delta}{N}}. \tag{29}$$

Similarly, we have with at least probability $1 - \frac{\delta}{3}$, for all $\nu \in \mathcal{V}$,

$$|\frac{1}{N_0} \sum_{j=1}^{N_0} \nu(s_{0,j}) - \mathbb{E}_\rho[\nu(s)]| \leq B_{\nu,\alpha}\sqrt{\frac{2\log 6|\mathcal{V}|/\delta}{N_0}}. \tag{30}$$

Therefore, with at least probability $1 - \frac{2}{3}\delta$ we have

$$T_2, T_4 \leq \alpha(B_{f,\alpha} + \frac{B_{w,\alpha}B_{e,\alpha}}{\alpha} + B_{e,\alpha}^2)\sqrt{\frac{2\log 6|\mathcal{V}|/\delta}{N}} + B_{\nu,\alpha}\sqrt{\frac{2\log 6\mathcal{V}|/\delta}{N_0}}. \tag{31}$$

● **Upper bounded of $T_1, T_5$**
For all $\nu \in \mathcal{V}$, we can express the difference of $L(w_\nu^*, \nu)$ and $L(\widehat{w}_\nu, \nu)$ by

$$|L(w_\nu^*, \nu) - L(\widehat{w}_\nu, \nu)| = \alpha\mathbb{E}_\mu\left[ h\left( \frac{1}{\alpha}e_\nu(s,a) \right) - h\left( \frac{1}{\alpha}\widehat{e}_\nu(s,a) \right) \right]$$
$$+ \mathbb{E}_\mu\left[ \mathbf{1}_{w_\nu^*(s,a)>0}e_\nu(s,a)^2 - \mathbf{1}_{\widehat{w}_\nu(s,a)>0}\widehat{e}_\nu(s,a)^2 \right]. \tag{32}$$

We bound the error of the squared terms first. By Lemma 3, one has

$$|\mu(s,a)[e_\nu(s,a) - \widehat{e}_\nu(s,a)]| = \mu(s,a) \sum_{s'} |\widehat{P}(s'|s,a) - P(s'|s,a)| \leq \epsilon_m, \tag{33}$$

then we can deduce that $\widehat{e}_\nu(s,a) < \alpha f'(0)$ given that $e_\nu(s,a) < \alpha f'(0) - \epsilon_m/\mu(s,a)$, and $\widehat{e}_\nu(s,a) > \alpha f'(0)$ given that $e_\nu(s,a) > \alpha f'(0) + \epsilon_m/\mu(s,a)$, due to the assumption that $e_\nu(s,a) \notin (\alpha f'(0) - \epsilon_m/\mu(s,a), \alpha f'(0) + \epsilon_m/\mu(s,a))$ for all $\nu \in V$. So

$$\mathbb{E}_\mu\left[ \mathbf{1}_{w_\nu^*(s,a)>0}e_\nu(s,a)^2 - \mathbf{1}_{\widehat{w}_\nu(s,a)>0}\widehat{e}_\nu(s,a)^2 \right]$$
$$= \mathbb{E}_\mu\left[ \mathbf{1}_{e_\nu(s,a)>\alpha f'(0)}e_\nu(s,a)^2 - \mathbf{1}_{\widehat{e}_\nu(s,a)>\alpha f'(0)}\widehat{e}_\nu(s,a)^2 \right]$$
$$= \mathbb{E}_\mu \mathbf{1}_{e_\nu(s,a)>\alpha f'(0)}\left[ e_\nu(s,a)^2 - \widehat{e}_\nu(s,a)^2 \right] \leq \mathbb{E}_\mu\left[ (e_\nu(s,a) + \widehat{e}_\nu(s,a))(e_\nu(s,a) - \widehat{e}_\nu(s,a)) \right]$$
$$\leq 2B_{e,\alpha}\mathbb{E}_\mu|e_\nu(s,a) - \widehat{e}_\nu(s,a)| \leq 2B_{e,\alpha}B_{\nu,\alpha}\mathbb{E}_\mu \sum_{s'} |\widehat{P}(s'|s,a) - P(s'|s,a)|. \tag{34}$$

To bound the rest error in (32), from (Lee et al., 2021) we have

$$h'(x) = - \underbrace{f'\left( (f')^{-1}(x) \right)}_{\text{(identity function)}} \left( (f')^{-1} \right)'(x) + \left( (f')^{-1} \right)'(x) \cdot x + (f')^{-1}(x)$$
$$= -x \cdot \left( (f')^{-1} \right)'(x) + \left( (f')^{-1} \right)'(x) \cdot x + (f')^{-1}(x)$$
$$= (f')^{-1}(x).$$

Since $(f')^{-1}(x)$ is upper bounded by $B_{w,\alpha}$ by (ii) of Assumption 3, $h'(x)$ is upper bounded by $B_{w,\alpha}$ too, which implies that $h(x)$ is Lipschitz continuous with a Lipschitz constant $B_{w,\alpha}$, thus

$$\alpha\mathbb{E}_\mu\left[ h\left( \frac{1}{\alpha}e_\nu(s,a) \right) - h\left( \frac{1}{\alpha}\widehat{e}_\nu(s,a) \right) \right] \leq B_{w,\alpha}\mathbb{E}_\mu|e_\nu(s,a) - \widehat{e}_\nu(s,a)|$$
$$\leq B_{w,\alpha}\mathbb{E}_\mu \sum_{s'} |\widehat{P}(s'|s,a) - P(s'|s,a)|\nu(s') \leq B_{w,\alpha}B_{\nu,\alpha}\mathbb{E}_\mu \sum_{s'} |\widehat{P}(s'|s,a) - P(s'|s,a)| \tag{35}$$

Combining (34),(35) and Lemma 3 yields that with probability at least $1 - \frac{\delta}{3}$,

$$|L(w_\nu^*, \nu) - L(\widehat{w}_\nu, \nu)| \leq 2(B_{w,\alpha} + 2B_{e,\alpha})B_{\nu,\alpha}\sqrt{\frac{\log 3|\mathcal{P}|/\delta}{N_m}} \tag{36}$$

By (24), (31) and (36) together we obtain Lemma 6. $\qquad\square$

**Lemma 7.** *The solution $\widehat{\nu}$ to $\widehat{L}_\mathcal{D}(\widehat{w}_\nu, \nu)$ and the optimal $\nu^*$ satisfy*

$$\|w_{\widehat{\nu}}^* - w_{\nu^*}^*\|_{2,\mu} \leq \frac{B_{f',\alpha}}{\alpha}\sqrt{1-\gamma}\sqrt{\epsilon_{N,N_0,N_m}}. \tag{37}$$

*Proof.* By the definition of $L(w_\nu^*, \nu)$,

$$L(w_\nu^*, \nu) = (1-\gamma)\mathbb{E}_{s\sim\rho(s)}[\nu(s)] + \mathbb{E}_{(s,a)\sim\mu}\left[\alpha h\left(\frac{1}{\alpha}e_\nu(s,a)\right) + \mathbf{1}_{e_\nu(s,a)>\alpha f'(0)}e_\nu(s,a)]^2\right]. \tag{38}$$

Now we denote $\mathcal{Z}_1 = \{(s,a) : e_{\nu^*}(s,a) > \alpha f'(0)\}$, $\mathcal{Z}_2 = \{(s,a) : e_{\nu^*}(s,a) \leq \alpha f'(0)\}.$, $\mathcal{S}_1 = \{s : \sum_a w_{\nu^*}^*(s,a) > 0\}$ and $\mathcal{S}_2 = \{s : \sum_a w_{\nu^*}^*(s,a) = 0\}$.

For the learned $\widehat{\nu}$, there exist three cases:
Case (i):

$$e_{\widehat{\nu}}(s,a) \begin{cases} > \alpha f'(0), \forall (s,a) \in \mathcal{Z}_1, \\ \leq \alpha f'(0), \forall (s,a) \in \mathcal{Z}_2 \end{cases} \tag{39}$$

At this time,

$$\begin{aligned} L(w_{\widehat{\nu}}^*, \widehat{\nu}) = {}&= (1-\gamma)\sum_{s\in\mathcal{S}_1}\rho(s)\widehat{\nu}(s) + \sum_{(s,a)\in\mathcal{Z}_1}\mu(s,a)\left(\alpha h(\frac{1}{\alpha}e_{\widehat{\nu}}(s,a)) + e_{\widehat{\nu}}^2(s,a)\right) \\ &+ (1-\gamma)\sum_{s\in\mathcal{S}_2}\rho(s)\widehat{\nu}(s) + \sum_{(s,a)\in\mathcal{Z}_2}\mu(s,a)\left(\alpha h(\frac{1}{\alpha}e_{\widehat{\nu}}(s,a))\right) \\ &:= L_{\mathcal{Z}_1}(w_{\widehat{\nu}}^*, \widehat{\nu}) + L_{\mathcal{Z}_2}(w_{\widehat{\nu}}^*, \widehat{\nu}). \end{aligned} \tag{40}$$

Since $\rho(s) = 0$ on $\mathcal{S}_2$, then $L_{\mathcal{Z}_2}(w_{\widehat{\nu}}^*, \widehat{\nu}) = -\alpha f(0)\sum_{(s,a)\in\mathcal{Z}_2}\mu(s,a) = L_{\mathcal{Z}_2}(w_{\nu^*}^*, \nu^*)$. Therefore, we have

$$L(w_{\widehat{\nu}}^*, \widehat{\nu}) - L(w_{\nu^*}^*, \nu^*) = L_{\mathcal{Z}_1}(w_{\widehat{\nu}}^*, \widehat{\nu}) - L_{\mathcal{Z}_1}(w_{\nu^*}^*, \nu^*) \leq \epsilon_{N,N_0,N_m}. \tag{41}$$

Combining the fact that $h''(x) \geq 0$ for all $x$ from Proposition 2 in (Lee et al., 2021) and the 2-strongly-convexity of $x^2$, we know that $L(w_\nu^*, \nu)$ is $2(1-\gamma)$-strongly-convex with respect to $\nu$ and $\|\cdot\|_{2,\mu}$ on $\mathcal{Z}_1$, yielding

$$\sqrt{\sum_{(s,a)\in\mathcal{Z}_1}(\widehat{\nu}(s,a) - \nu^*(s,a))^2} \leq \sqrt{\frac{L_{\mathcal{Z}_1}(w_{\widehat{\nu}}^*, \widehat{\nu}) - L_{\mathcal{Z}_2}(w_{\nu^*}^*, \nu^*)}{(1-\gamma)}} \leq \sqrt{\frac{\epsilon_{N,N_0,N_m}}{(1-\gamma)}}. \tag{42}$$

Then on $\mathcal{Z}_1$, we have

$$\begin{aligned} \|w_{\widehat{\nu}}^* - w_{\nu^*}^*\|_{2,\mu} &\leq \frac{B_{f',\alpha}}{\alpha}\sqrt{\mathbb{E}_\mu[e_{\widehat{\nu}}(s,a) - e_{\nu^*}(s,a)]^2} \\ &= \frac{B_{f',\alpha}}{\alpha}\|e_{\widehat{\nu}}(s,a) - e_{\nu^*}(s,a)\|_{2,\mu} \leq \frac{B_{f',\alpha}}{\alpha}(1-\gamma)\|\widehat{\nu}(s,a) - \nu^*(s,a)\|_{2,\mu} \\ &\leq \frac{B_{f',\alpha}}{\alpha}\sqrt{1-\gamma}\sqrt{\epsilon_{N,N_0,N_m}}. \end{aligned} \tag{43}$$

Combining the fact that $w_{\widehat{\nu}}^*(s,a) = w_{\nu^*}^*(s,a) = 0$ on $\mathcal{Z}_2$, Lemma 7 holds true for all $(s,a)$.

Case (ii): there exists some $(\bar{s}, \bar{a}) \in \mathcal{Z}_1$ such that $e_{\widehat{\nu}}(\bar{s}, \bar{a}) \leq \alpha f'(0)$. By the optimality of $L(w_{\nu^*}^*, \nu^*)$ and $L(w_{\nu^*}^*, \nu^*) - \mathbb{E}_\mu \mathbf{1}_{e_{\nu^*}>\alpha f'(0)}e_{\nu^*}^2$, the density ratio drop from $w_{\nu^*}^*(\bar{s}, \bar{a})$ to 0 must

be compensated by some other $(\tilde{s}, \tilde{a})$, such that $w^*_{\widehat{\nu}}(\tilde{s}, \tilde{a}) > 0$ and its positive contribution to $L(w^*_{\nu^*}, \nu^*) - \mathbb{E}_\mu \mathbf{1}_{e_{\nu^*} > \alpha f'(0)} e^2_{\nu^*}$ satisfies

$$(1 - \gamma)\rho(\tilde{s}))[\widehat{\nu}(\tilde{s})] + \alpha\mu(\tilde{s}, \tilde{a})h\left(\frac{1}{\alpha}e_{\widehat{\nu}}(\tilde{s}, \tilde{a})\right) - (1 - \gamma)\rho(\tilde{s}))[\nu^*(\tilde{s})] - \alpha\mu(\tilde{s}, \tilde{a})h\left(\frac{1}{\alpha}e_{\nu^*}(\tilde{s}, \tilde{a})\right) \geq O(\Delta), \tag{44}$$

where we use $O(\Delta)$ to represent some constant independent of $\epsilon_{N, N_0, N_m}$. This implies that the change from $\nu^*(\tilde{s})$ to $\widehat{\nu}(\tilde{s})$ will be larger than some constant $O(\Delta)$by Assumption 3. Then by the optimality of $L(w^*_{\nu^*}, \nu^*)$,

$$L_(w^*_{\widehat{\nu}}, \widehat{\nu}) - L_(w^*_{\nu^*}, \nu^*) \geq e^2_{\widehat{\nu}}(\tilde{s}, \tilde{a}) \geq O(\Delta), \tag{45}$$

this is contradictory to Lemma 6 So it is impossible for case (ii) to occur.

Case (iii): there exists some $(\bar{s}, \bar{a}) \in \mathcal{Z}_2$ such that $e_{\widehat{\nu}}(\bar{s}, \bar{a}) > \alpha f'(0)$. Then by the optimality of $L(w^*_{\nu^*}, \nu^*)$,

$$L(w^*_{\widehat{\nu}}, \widehat{\nu}) \geq L(w^*_{\nu^*}, \nu^*) + e^2_{\widehat{\nu}}(\bar{s}, \bar{a}). \tag{46}$$

Combining Lemma 6 we have

$$e^2_{\widehat{\nu}}(\bar{s}, \bar{a}) \leq \epsilon_{N, N_0, N_m}, \tag{47}$$

then for small enough $\epsilon_{N, N_0, N_m}$,

$$e_{\widehat{\nu}}(\bar{s}, \bar{a}) - \alpha f'(0) > |\alpha f'(0)/2| := O(\Delta) \tag{48}$$

Then by Assumption (ii) of 3, we have

$$w^*_{e_{\widehat{\nu}}}(\bar{s}, \bar{a}) \geq O(\Delta). \tag{49}$$

The growth from $w^*_{\nu^*}(\bar{s}, \bar{a}) = 0$ to $w^*_{\widehat{\nu}}(\bar{s}, \bar{a})$ necessarily accompanies the decrease from $w^*_{\nu^*}(\tilde{s}, \tilde{a})$ to $w^*_{\widehat{\nu}}(\tilde{s}, \tilde{a})$ for some $(\tilde{s}, \tilde{a}) \in \mathcal{Z}_1$, then from (ii) in Assumption 3, the change from $e_{\nu^*}(\tilde{s}, \tilde{a})$ to $e_{\widehat{\nu}}(\tilde{s}, \tilde{a})$ should be larger than some constant $O(\Delta)$. At this time,

$$L(w^*_{\widehat{\nu}}, \widehat{\nu}) - L(w^*_{\nu^*}, \nu^*) \geq e^2_{\widehat{\nu}}(\tilde{s}, \tilde{a}) \geq O(\Delta), \tag{50}$$

which is contradictory to Lemma 6 again. So case (iii) is also impossible to occur.

Considering the overall cases, it can be concluded that Lemma 7 is valid.

$\square$

**Lemma 8.**

$$\|\widehat{w}_{\widehat{\nu}} - w^*_{\nu^*}\|_{2,\mu} \leq \frac{B_{f',\alpha}}{\alpha}(\sqrt{2B_{e,\alpha}} + \sqrt{1 - \gamma})\sqrt{\epsilon_{N, N_0, N_m}}. \tag{51}$$

*Proof.* We decompose the objective by

$$\|\widehat{w}_{\widehat{\nu}} - w^*_{\nu^*}\|_{2,\mu} = \|\widehat{w}_{\widehat{\nu}} - w^*_{\widehat{\nu}}\|_{2,\mu} + \|w^*_{\widehat{\nu}} - w^*_{\nu^*}\|_{2,\mu}. \tag{52}$$

It is easy to prove that $\max\left(0, (f')^{-1}(x)\right)$ is also $B_{f',\alpha}$-continuous through (ii) of Assumption 3, then

$$\begin{aligned}
\|\widehat{w}_{\widehat{\nu}} - w^*_{\widehat{\nu}}\|_{2,\mu} &\leq \frac{B_{f',\alpha}}{\alpha}\sqrt{\mathbb{E}_\mu[\widehat{e}_{\widehat{\nu}}(s, a) - e_{\widehat{\nu}}(s, a)]^2} \\
&\leq \sqrt{2B_{e,\alpha}}\frac{B_{f',\alpha}}{\alpha}\sqrt{\mathbb{E}_\mu|\widehat{e}_{\widehat{\nu}}(s, a) - e_{\widehat{\nu}}(s, a)|} \leq \sqrt{2B_{e,\alpha}}\frac{B_{f',\alpha}}{\alpha}\left(\frac{4\log 3|\mathcal{P}|/\delta}{N}\right)^{1/4} \\
&\leq \sqrt{2B_{e,\alpha}}\frac{B_{f',\alpha}}{\alpha}\epsilon_{N, N_0, N_m}, \tag{53}
\end{aligned}$$

where penultimate line holds due to (35) and (21).

Combine (53) and Lemma 7 together yields Lemma 8.

$\square$

*Proof of Theorem 4.* Notice that $\widehat{w}_{\widehat{\nu}} = \widehat{w}_\alpha$ and $w^*_{\nu^*} = w^*_\alpha$, then by Lemma 6 and Lemma 8, we can obtain the second inequality of Theorem 4, where

$$\mathcal{E}_{N,N_0,N_m,\alpha} = \frac{B_{f',\alpha}}{\alpha}(\sqrt{2B_{e,\alpha}} + \sqrt{1-\gamma})\sqrt{\epsilon_{N,N_0,N_m}}. \tag{54}$$

The first inequality holds since

$$\|\widehat{d}_{\widehat{w}_\alpha} - d^*_\alpha\|_1 = \|\widehat{w}_\alpha - w^*_\alpha\|_{1,\mu} \leq \|\widehat{w}_\alpha - w^*_\alpha\|_{2,\mu}. \tag{55}$$

Then we finish the proof. □

### A.3 PROOF OF THEOREM 6

*Proof of Theorem 6.* Similar to the proof in Corollary 1 in (Zhan et al., 2022), we divide the proof into two steps. We first show that $d^*_0 - d^*_{\alpha_\epsilon} \leq \frac{\epsilon}{2}$ and we then bound $d^*_{\alpha_\epsilon} - \widehat{d}_{\widehat{w}_{\alpha_\epsilon}}$ by utilizing Theorem 4.

●: **Bounding** $d^*_0 - d^*_{\alpha_\epsilon}$. Notice that $d^*_{\alpha_\epsilon}$ is the stationary distribution of $\pi^*_{\alpha_\epsilon}$ and $\pi^*_{\alpha_\epsilon}$ is the solution to the regularized problem, then

$$\mathbb{E}_{(s,a)\sim d^*_{\alpha_\epsilon}}[r(s,a)] - \alpha\mathbb{E}_{(s,a)\sim\mu}\left[f\left(w^*_{\alpha_\epsilon}(s,a)\right)\right] \geq \mathbb{E}_{(s,a)\sim d^*_0}[r(s,a)] - \alpha\mathbb{E}_{(s,a)\sim\mu}\left[f\left(w^*_0(s,a)\right)\right],$$

which implies

$$
\begin{aligned}
J\left(\pi^*_0\right) - J\left(\pi^*_{\alpha_\epsilon}\right) &= \mathbb{E}_{(s,a)\sim d^*_0}[r(s,a)] - \mathbb{E}_{(s,a)\sim d^*_{\alpha_\epsilon}}[r(s,a)] \\
&\leq \alpha\mathbb{E}_{(s,a)\sim\mu}\left[f\left(w^*_0(s,a)\right)\right] - \alpha\mathbb{E}_{(s,a)\sim\mu}\left[f\left(w^*_{\alpha_\epsilon}(s,a)\right)\right] \\
&\leq \alpha\mathbb{E}_{(s,a)\sim\mu}\left[f\left(w^*_0(s,a)\right)\right] \\
&\leq \alpha B_{f,0}.
\end{aligned}
\tag{56}
$$

Combining the assumption that $r > 0$ and the choice of $\alpha_\epsilon$, we have

$$\|d^*_0 - d^*_{\alpha_\epsilon}\| \leq \frac{\epsilon}{2}. \tag{57}$$

●: **Bounding** $d^*_{\alpha_\epsilon} - \widehat{d}_{\widehat{w}_{\alpha_\epsilon}}$. Using Theorem 4, we know that if $N, N_0, N_m$ satisfies the following conditions,

$$N \geq C_1 \left(\frac{B^2_{f,0}B^4_{f',\alpha}r^2}{\epsilon^2}(2B_{e,\alpha}+1-\gamma)^2(\frac{B^2_\nu}{\epsilon^2} + \frac{2B_{f,0}B_{w,\alpha}B_{e,\alpha}r}{\epsilon} + B^2_{e,\alpha})^2\right)\log\frac{6|\mathcal{V}|}{\delta},$$

$$N_m \geq C_2 \left(\frac{B^2_{f,0}B^4_{f',\alpha}r^2}{\epsilon^2}(2B_{e,\alpha}+1-\gamma)^2(2B_{w,\alpha}B_{\nu,\alpha}+4B_{e,\alpha}B_{\nu,\alpha})^2\right)\cdot\log\frac{3|\mathcal{P}|}{\delta}$$

$$N_0 \geq C_3 \left(\frac{B^2_{f,0}B^4_{f',\alpha}r^2}{\epsilon^2}(2B_{e,\alpha}+1-\gamma)^2B^2_{\nu,\alpha}\right)\log\frac{6|\mathcal{V}|}{\delta}, \tag{58}$$

then with at least probability $1 - \delta$,

$$\|d^*_{\alpha_\epsilon} - \widehat{d}_{\alpha_\epsilon}\| \leq \frac{\epsilon}{2}. \tag{59}$$

Using (59) and (57), we concludes that

$$\|d^*_0 - \widehat{d}_{\alpha_\epsilon}\|_1 \leq \epsilon \tag{60}$$

holds with at least probability $1 - \delta$. This finishes our proof. □

### A.4 PROOF OF THEOREM 7

Two lemmas are first introduced as follows:

**Lemma 9** (Theorem 1 in (Zhan et al., 2022))**.** *Suppose that Assumption 1 -4 hold, $\pi_{\widehat{w}_\alpha}$ is the policy induced by the output of Algorithm 1, then*

$$J(\pi_\alpha^*) - J(\pi_{\widehat{w}_\alpha}) \leq \frac{1}{1-\gamma}\mathbb{E}_{d_\alpha^*}[\|\pi_\alpha^*(\cdot|s) - \pi_{\widehat{w}_\alpha}(\cdot|s)\|_1] \leq \frac{2}{1-\gamma}\|\widehat{w}_\alpha - w_\alpha^*\|_{2,\mu}. \tag{61}$$

**Lemma 10.** *(Convergence of MLE for behavior cloning (Rajaraman et al., 2020)) For all $w$, Given a realizable policy class $\Pi_w$ that contains the true policy $\pi_w$ and the dataset $\mathcal{D}_w = \{(s_i, a_i)\}$ with $s_i \in d_w(s)$, $a_i \in \pi_w(a_i|s_i)$, let $\widehat{\pi}_w$ be*

$$\widehat{\pi}_w = \arg\max_{\tilde{\pi}_w \in \Pi_w} \sum_{i=1}^{N} w(s_i, a_i) \log \tilde{\pi}_w(a_i|s_i), \tag{62}$$

*then with probability at least $1 - \delta$,*

$$\mathbb{E}_{d_w(s)}\|\widehat{\pi}_w - \pi_w\|_1 \leq \sqrt{\frac{\log|\Pi_w||W|/\delta}{N}} \tag{63}$$

*Proof of Theorem 7.* By performance difference inequality we have

$$J(\pi_\alpha^*) - J(\widehat{\pi}_\alpha^{\text{osd-bc}}) \leq \frac{1}{1-\gamma}\mathbb{E}_{d_\alpha^*}[\|\pi_\alpha^* - \widehat{\pi}_\alpha^{\text{osd-bc}}\|_1]$$

$$\leq \frac{1}{1-\gamma}\mathbb{E}_{d_\alpha^*}[\|\pi_\alpha^* - \pi_{\widehat{w}_\alpha}\|_1] + + \frac{1}{1-\gamma}\mathbb{E}_{d_\alpha^*}[\|\pi_{\widehat{w}_\alpha} - \widehat{\pi}_\alpha^{\text{osd-bc}}\|_1]. \tag{64}$$

The first term can be bounded by $\frac{2}{1-\gamma}\|\widehat{w}_\alpha - w_\alpha^*\|_{2,\mu}$ by Lemma 9. The second term satisfies

$$\frac{1}{1-\gamma}\mathbb{E}_{d_\alpha^*}[\|\pi_{\widehat{w}_\alpha} - \widehat{\pi}_{\widehat{w}_\alpha}^{\text{osd-bc}}\|_1] \leq \frac{1}{1-\gamma}\mathbb{E}_{d_{\widehat{w}_\alpha}}[\|\pi_{\widehat{w}_\alpha} - \widehat{\pi}_{\widehat{w}_\alpha}^{\text{osd-bc}}\|_1] + \frac{1}{1-\gamma}\mathbb{E}_{|d_\alpha^* - d_{\widehat{w}_\alpha}|}[\|\pi_{\widehat{w}_\alpha} - \widehat{\pi}_{\widehat{w}_\alpha}^{\text{osd-bc}}\|_1]$$

$$\leq \frac{2}{(1-\gamma)^2}\sqrt{\frac{\ln|\Pi_w||W|/\delta}{N}} + \frac{1}{1-\gamma}\||d_\alpha^* - d_{\widehat{w}_\alpha}\||_1 \tag{65}$$

$$\leq \frac{2}{(1-\gamma)^2}\sqrt{\frac{\ln|\Pi_w||W|/\delta}{N}} + \frac{1}{1-\gamma}\sqrt{\mathcal{E}_{N,N_0,N_m,\alpha}}, \tag{66}$$

$$= \frac{2}{(1-\gamma)^2}\sqrt{\frac{\ln|\Pi_\nu||\mathcal{V}|/\delta}{N}} + \frac{1}{1-\gamma}\mathcal{E}_{N,N_0,N_m,\alpha}$$

$$= \frac{1}{1-\gamma}\left(\mathcal{E}_{N,N_0,N_m,\alpha} + \frac{2}{1-\gamma}\mathcal{E}_N'\right)$$

where (65) comes from Lemma 10 and Theorem 4, respectively.

Putting (64) -(66) together finishes our proof. $\qquad\square$

## A.5   PROOF OF THEOREM 9

Recall the following minimax offline RL formulation with relative pessimism proposed in (Cheng et al.),

$$\widehat{\pi}^* \in \underset{\pi \in \Pi}{\text{argmax}}\mathcal{L}_\mu(\pi, f^\pi) \tag{67}$$

$$\text{s.t. } f^\pi \in \underset{f \in \mathcal{F}}{\text{argmin}}\mathcal{L}_\mu(\pi, f) + \beta\mathcal{E}_\mu(\pi, f) \tag{68}$$

where $\beta \geq 0$ is a hyperparamter, and

$$\mathcal{L}_\mu(\pi, f) := \mathbb{E}_\mu[f(s, \pi) - f(s, a)] \tag{69}$$

$$\mathcal{E}_\mu(\pi, f) := \mathbb{E}_\mu\left[((f - \mathcal{T}^\pi f)(s, a))^2\right]. \tag{70}$$

It is proved that

**Lemma 11** (Proposition 3 in (Cheng et al.))**.** *If $Q^\pi \in \mathcal{F}$, and $\mu(a|s) \in \Pi$, then for any $\beta \geq 0$, $J(\widehat{\pi}^*) \geq J(\mu)$.*

*Proof of Theorem 9.* By replacing $\mu$ in this formulation with $d_{\widehat{w}_\alpha}$, Theorem 9 can be easily obtained by combining Lemma 11, Lemma 9 and Theorem 4 together. $\qquad\square$

A.6 Proof of Theorem 10

The formal version of Theorem 10 is as follows:

**Theorem 11.** *Suppose that Assumptions 1-4 hold. Let $\widehat{\pi}^{osd\text{-}CQL}$ be the policy obtained by osd-CQL. Then with at least $1 - 2\delta$, $\widehat{\pi}^{osd\text{-}CQL}$ satisfies*

$$J(\widehat{\pi}^{osd\text{-}CQL}) \geq J(\pi_\alpha^*) - \frac{2}{1-\gamma}\mathcal{E}_{N,N_0,N_m,\alpha} - \zeta, \tag{71}$$

*with high probability $1 - 2\delta$, where $\zeta$ is given by*

$$\zeta = 2\left(\frac{C_{r,\delta}}{1-\gamma} + \frac{\gamma C_{T,\delta}}{(1-\gamma)^2}\right)\mathbb{E}_{s\sim d_{\widehat{M}}^{\widehat{\pi}^{osd\text{-}CQL}}}\left[\frac{\sqrt{|\mathcal{A}|}}{\sqrt{|N|}}\sqrt{D_{CQL}\left(\widehat{\pi}^{osd\text{-}CQL}, \pi_{\widehat{w}_\alpha}\right)(s) + 1}\right]$$

$$- \alpha_{cql}\frac{1}{1-\gamma}\mathbb{E}_{s\sim d_{\widehat{M}}^{\widehat{\pi}^{osd\text{-}CQL}}}[D_{CQL}\left(\widehat{\pi}^{osd\text{-}CQL}, \pi_{\widehat{w}_\alpha}\right)(s)]. \tag{72}$$

*$C_{r,\delta}$ and $C_{T,\delta}$ are constants defined in (Kumar et al., 2020), $\widehat{M}$ is the empirical dynamic kernel, $D_{CQL}(\cdot, \cdot)$ is the CQL distance between two policies.*

The first term and the second term of $\zeta$ are called "sampling error" and "positive term" in Theorem 10.

*Proof.* Since we conduct CQL on $\mathcal{D}_{osd}$, this means that the behavior policy is $\pi_{\widehat{w}_\alpha}$. Then by Theorem 3.6 in (Kumar et al., 2020), it is straightforward to derive that with probability at least $1 - \delta$,

$$J(\widehat{\pi}^{osd\text{-}CQL}) \geq J(\pi_{\widehat{w}_\alpha}) - \zeta. \tag{73}$$

Then combining (73) with Lemma 9, we finish our proof. □

# B EXPERIMENTAL CONFIGURATION FOR FINITE MDPS

**OSD-DICE for Finite MDPs** For finite MDPs, we can reformulate the expected objective of OSD-DICE into vector-matrix form:

$$\min_\nu \max_{w\geq 0} L_\alpha(w, \nu) = (1-\gamma)\rho^\top\nu - \frac{\alpha}{2}(w-1)^\top D(w-1) + w^\top De_\nu + \beta(\mathbf{1}_{w>0}e_\nu)^\top D(\mathbf{1}_{w>0}e_\nu) \tag{74}$$

where $\nu \in \mathbb{R}^{|S|}$ is a $|S|$-dimensional vector, $w \in \mathbb{R}^{|S||A|}$ is a $|S||A|$-dimensional vector, and $R \in \mathbb{R}^{|S||A|}$ is a $|S||A|$-dimensional reward vector, and $D = \text{diag}\left(d^{\pi_\mu}\right) \in \mathbb{R}^{|S||A|\times|S||A|}$ is the diagonal matrix induced by $\mu$, $\beta$ is used to control the degree of the squared term and is set as $0.01$. Moreover, we choose the $\mathcal{X}^2$-divergence by $f(x) = \frac{1}{2}(x-1)^2$ for brevity. Let $\mathcal{T} \in \mathbb{R}^{|S||A|\times|S|}$ and $\mathcal{B} \in \mathbb{R}^{|S||A|\times|S|}$ be the matrices satisfying the following equations:

$$\mathcal{T}\nu \in \mathbb{R}^{|S||A|} \quad \text{s.t.} \quad (\mathcal{T}\nu)((s,a)) = \sum_{s'} P\left(s' \mid s, a\right)\nu\left(s'\right)$$

$$\mathcal{B}\nu \in \mathbb{R}^{|S||A|} \quad \text{s.t.} \quad (\mathcal{B}\nu)((s,a)) = \nu(s) \tag{75}$$

So $e_\nu = R + \gamma\mathcal{T}\nu - \mathcal{B}\nu$. Then, by substituting the closed-form solution of the inner maximization $w_\nu^* = \max\left(0, \frac{1}{\alpha}e_\nu + 1\right)$ into $L_\alpha(w, \nu)$ and considering $(f')^{-1}(x) = x + 1$, we can get:

$$\min_\nu L\left(w_\nu^*, \nu\right) = L(\nu) := (1-\gamma)\rho^\top\nu - \frac{\alpha}{2}\left(w_\nu^* - 1\right)D\left(w_\nu^* - 1\right) + w_\nu^{*\top}De_\nu + \beta(\mathbf{1}_{w_\nu^*>0}e_\nu)^\top D(\mathbf{1}_{w_\nu^*>0}e_\nu) \tag{76}$$

In OSD-DICE, we use some approximator $\tilde{P}$ in the function class $\mathcal{P}$ to approximate $P$ and obtain $\tilde{e}_\nu$. We then perform Newton's method to solve the optimization problem with the gradients and the Hessian of the objective with respect to $\nu$. We can derive the gradient and the Hessian with the

following steps, and then perform the iteration process to compute an optimal $\nu^*$:

$$m = \mathbb{I}\left(\frac{1}{\alpha}\tilde{e}_\nu + 1 \geq 0\right)$$

$$w_\nu^* = \left(\frac{1}{\alpha}\tilde{e}_\nu + 1\right) \odot m$$

$$J = \frac{\partial w_\nu^*}{\partial \nu} = \frac{1}{\alpha}(\gamma\mathcal{T} - \mathcal{B}) \odot m$$

$$g = \frac{\partial L(\nu)}{\partial \nu} = (1-\gamma)\rho - \alpha J^\top D\left(w_\nu^* - 1\right) + J^\top D\tilde{e}_\nu + (\gamma\mathcal{T} - \mathcal{B})^\top Dw_\nu^* + 2\beta(\gamma\mathcal{T} - \mathcal{B})^\top D\tilde{e}_\nu \mathbf{1}_{\tilde{w}_\nu > 0}$$

$$H = \frac{\partial^2 L(\nu)}{\partial \nu^2} = -\alpha J^\top DJ + J^\top D(\gamma\mathcal{T} - \mathcal{B}) + (\gamma\mathcal{T} - \mathcal{B})^\top DJ + 2\beta(\gamma\mathcal{T} - \mathcal{B})^\top D(\gamma\mathcal{T} - \mathcal{B})\mathbf{1}_{\tilde{w}_\nu > 0}$$

$$(77)$$

Note that $\tilde{P}$ is designed to be selected from a realizable function class $\mathcal{P} := \{P : P = \theta_0 P^* + \theta_1 P_1 + \theta_2 P_2 + \theta_3 P_3\}$ with $P_1, P_2, P_3$ being some fixed stochastic matrix and $\theta := (\theta_0, \theta_2, \theta_3, \theta_4)$ being learned. Then $\tilde{P}$ can be much more accurate than the empirical $\widehat{P}$ induced by the single-transition estimation used in OptiDICE (Lee et al., 2021).

**MDP generation** In alignment with the experimental settings in (Lee et al., 2021), we employ randomly generated MDPs with a state space size $|S| = 60$, an action space size $|A| = 8$, a discount factor of $\gamma = 0.95$, and a deterministic initial state distribution, i.e., $\rho(s) = 1$ for a fixed $s = s_0$. The transition model has a connectivity of 8, meaning that for every state-action pair $(s, a)$, non-zero transition probabilities are allocated to 8 distinct states $(s_1', \cdots, s_8')$. The transition probabilities are generated randomly using a Dirichlet distribution $[p(s_1' \mid s, a), p(s_2' \mid s, a), p(s_3' \mid s, a), p(s_4' \mid s, a), p(s_5' \mid s, a), p(s_6' \mid s, a), p(s_7' \mid s, a), p(s_8' \mid s, a)] \sim \text{Dir}(1, 1, 1, 1, 1, 1, 1, 1)$. A reward of 1 is assigned to the state that minimizes the optimal state value at the initial state; other states are associated with zero rewards. This reward function design can be perceived as a method of specifying a target state that poses the greatest challenge in terms of accessibility from the initial state. The episode terminates once the agent successfully reaches the rewarding goal state.

**Task Descriptions** We perform an extensive empirical evaluation of the tabular OSD-DICE's efficacy and stability. This evaluation leverages a series of MDPs which are generated randomly and accompanied by varying numbers of trajectories and varying degrees of optimality of the data-collection optimality. The experiment is conducted in accordance with the protocols in (Lee et al., 2021). Our experimental setup involves 10,000 independent runs. In each run, we first randomly generate an MDP and then formulate a data-collection policy aligned with a predefined degree of optimality, $\zeta \in \{0.8, 0.9\}$. Following this, we collect $N$ trajectories (where $N \in \{200, 300, 400, 500, 600, 700, 800\}$) from the generated MDP and the associated data-collection policy $\pi_\mu$. Subsequently, the constructed data-collection policy and the accumulated trajectories are given to offline RL algorithm. We then evaluate the algorithm's performance, both in terms of the mean performance and the Conditional Value at Risk (CVaR) at the 5% threshold, which considers the worst 5% runs.

## C  EXPERIMENTAL CONFIGURATION FOR D4RL TASKS AND HYPERPARAMETER CONFIGURATION

**Task Descriptions** We use Maze2D, Gym-MuJoCo and Antmaze environments of D4RL benchmark (Fu et al., 2020) to evaluate our method in continuous control tasks. We summarize the descriptions of tasks in D4RL paper (Fu et al., 2020) as follows:

Maze2D is a navigation task in 2D state space, while the agent tries to reach a fixed goal location. By using priorly gathered trajectories, the goal of the agent is to find out a shortest path to reach the goal location. The complexity of the maze increases with the order of "maze2d-umaze", "maze2d-medium" and "maze2d-large".

Gym-MuJoCo locomotion domains involves three agents: halfcheetah, hopper, and walker2d. For each agent, four datasets are provided which correspond to behavior policies with different qualities: random, medium, medium-replay and medium-expert.

AntMaze domains consist of sparse-reward tasks and require "stitching" fragments of suboptimal trajectories traveling undirectedly to find a path from the start to the goal of the maze.

For Gym-MuJoCo locomotion domains, we use v2 datasets, and for the rest domains, we use v0 datasets.

**Implementation Details for OSD-DICE** In practice, we made two adaptation modifications for D4RL tasks compared to the theoretical algorithms: (i) we still adopt the single-transition estimator to approximate $e_\nu$, as the state and action spaces are continuous and the MDPs are approximately deterministic on these tasks, so the single-transition estimator is simple and reliable in this situation. (ii) we follow (Lee et al., 2021) to additionally add a normalization constraint $\sum_{s,a} w(s,a)\mu(s,a) = 1$ which does not not affect the correctness of the all the conclusions. Than (6) becomes

$$
\begin{aligned}
\min_{\nu \in \mathcal{V}} \widehat{L}_\alpha(\nu) = {} & (1-\gamma)\frac{1}{N_0}\sum_{j=1}^{N_0}[\nu(s_{0,j})] + \lambda + \frac{1}{N}\sum_{j=1}^{N}\left[-\alpha f\left(\max\left(0, (f')^{-1}\left(\frac{1}{\alpha}\widehat{e}_{\nu,\lambda}(s_j, a_j)\right)\right)\right)\right] \\
& + \frac{1}{N}\sum_{j=1}^{N}\left[\max\left(0, (f')^{-1}\left(\frac{1}{\alpha}\widehat{e}_{\nu,\lambda}(s_j, a_j)\right)\right)\widehat{e}_{\nu,\lambda}(s_j, a_j)\right] \\
& + \beta\frac{1}{N}\sum_{j=1}^{N}\mathbf{1}_{\widehat{w}_{\nu,\lambda}(s_j, a_j)>0}[\widehat{e}_{\nu,\lambda}(s_j, a_j)]^2,
\end{aligned}
\tag{78}
$$

where $\lambda$ is the Lagrange multiplier and $\widehat{e}_{\nu,\lambda} = \widehat{e}_\nu - \lambda$. We use $\nu_\theta$ network to represent $\nu$ and adopts fully-connected MLPs with two hidden layers and ReLU activations, where the number of hidden units on each layer is 256. For the optimization of $\nu$, we use stochastic gradient descent with Adam optimizer and its learning rate is 0.0003. The batch size is set to be 512. Following (Lee et al., 2021), we choose the soft version of $\mathcal{X}^2$-divergence by $f(x) = x\log x - x + 1$ if $0 < x < 1$ or $\frac{1}{2}(x-1)^2$ else. When combined with BC, we follow (Lee et al., 2021) to preprocess the dataset $\mathcal{D}$ by interpreting terminal states as absorbing states and using the absorbing-state implementation proposed in (Lee et al., 2021), and also standardize observations and rewards. When combined with CQL and TD3BC, we do not apply absorbing state and keep the dataset preprocessing method the same with CQL and TD3BC, respectively. The selection for $\alpha$ is listed in Table 2 and $\beta = 0.001$. After training $\nu$ and $\lambda$ for 500,000 steps, we fix the parameters and continue with the subsequent policy optimization. In the policy optimization phase, we set boundaries for the learned weights based on the histograms, resulting an interval of $[0.1, 10]$ for all tasks.

**Implementation Details for osd-BC** Our implementation of osd-BC builds upon the official code of (Lee et al., 2021). We use tanh-squashed normal distribution $\pi_\psi$ to represent the learning policy in osd-BC and $\pi_\psi$ adopts fully-connected MLPs with two hidden layers and ReLU activations, where the number of hidden units on each layer is equal to 256. We regularize the entropy of $\pi_\psi$ with learnable entropy regularization coefficients, where the target entropy are set to be the same as those in SAC. For the optimization of $\pi_\psi$, we use stochastic gradient descent with Adam optimizer and its learning rate is 0.0003. The batch size is set to be 512. Before training neural networks, we preprocess the dataset $\mathcal{D}$ by interpreting terminal states as absorbing states and using the absorbing-state implementation proposed in (Lee et al., 2021). We also standardize observations and rewards. $\pi_\psi$ is updated for 1000, 000 steps. For each task, we search the coefficient $\alpha$ within $\{0.0001, 0.001, 0.01, 0.1, 1\}$ which is described in Table 2:

**Implementation Details for osd-CQL and osd-TD3BC** We develop osd-CQL and osd-TD3BC based on codebase (Sun, 2023), and we keep the hyperparameters and implementation of CQL and TD3BC consistent with the codebase. For learning both the critic and actor, fully-connected MLPs with 3 hidden layers and ReLU activations are used, where the number of hidden units on each layer is 256. For CQL, the learning rates for critic and actor are 0.0003 and 0.0001 respectively, the conservative coefficient is 5, the actions samples number is 10, and the batch size is 256. For TD3BC, the learning rates for critic and actor are both 0.0003, and its conservative coefficient is 2.5, the exploration noise is 0.1, the policy noise is 0.2. We also implement osd-CQL based on `https://github.com/young-geng/CQL` and find it performs better on Antmaze domain, so we report the results of osd-CQL on Antmaze through this implementation. The selection for $\alpha$ is listed in Table 2, and $\beta = 0.001$. We update the critic and actor for 1000, 000 steps,

Table 2: Hyperaparameters for $\alpha$

| Task | osd-BC | osd-TD3BC | osd-CQL |
|---|---|---|---|
| maze2d-umaze | 0.001 | 0.001 | 0.001 |
| maze2d-medium | 0.001 | 0.001 | 0.001 |
| maze2d-large | 0.001 | 0.001 | 0.001 |
| hopper-random | 0.001 | 1.0 | 1.0 |
| hopper-medium | 1.0 | 1.0 | 1.0 |
| hopper-medium-replay | 1.0 | 1.0 | 1.0 |
| hopper-medium-expert | 0.01 | 1.0 | 1.0 |
| halfcheetah-random | 0.001 | 1.0 | 1.0 |
| halfcheetah-medium | 1.0 | 1.0 | 1.0 |
| halfcheetah-medium-replay | 1.0 | 1.0 | 1.0 |
| halfcheetah-medium-expert | 0.01 | 1.0 | 1.0 |
| walker2d-random | 1.0 | 1.0 | 1.0 |
| walker2d-medium | 1.0 | 1.0 | 1.0 |
| walker2d-medium-replay | 1.0 | 1.0 | 1.0 |
| walker2d-medium-expert | 0.01 | 1.0 | 1.0 |
| antmaze-umaze | 1.0 | 1.0 | 1.0 |
| antmaze-umaze-diverse | 1.0 | 1.0 | 1.0 |
| antmaze-medium-diverse | 1.0 | 1.0 | 1.0 |
| antmaze-medium-play | 0.01 | 1.0 | 1.0 |

**Evaluation Protocol and Baselines** We report the average undiscounted normalized return after the policy is trained for one million training steps, with 10 evaluation episodes per method. The BC-related baselines are developed using the official implementation of (Hong et al.). For CQL-based and TD3BC-based baselines, we re-implement top10%, AW and RW strategies on the same codebase (Sun, 2023) to achieve fair comparison. In particular, we adopt the same implementation of (Hong et al.) to implement top10%, AW and RW strategies on top of `https://github.com/young-geng/CQL` for Antmaze domain, to make a fair comparison for this domain. We also conduct hyperparameter selection for these baselines.

# D    ABLATION STUDY FOR $\alpha$ AND $\beta$

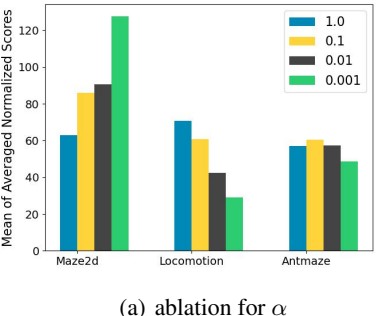

(a) ablation for $\alpha$

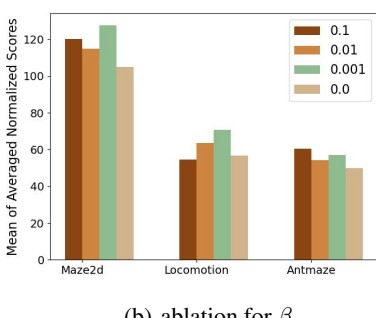

(b) ablation for $\beta$

Figure 2: Ablation Study for $\alpha$ and $\beta$

We conduct ablation studies for OSD-CQL to further assess the sensitivity of $\alpha$ and $\beta$ in Figure 2. Each bar in the histogram represents the average score of in the corresponding domain. It can be seen that OSD-CQL maintains good scores under different $\alpha$ on Maze2d and Antmaze, indicating that OSD-CQL is robust to $\alpha$ for these domains. It is worth mentioning that even though there exist better choices for $\alpha$ on Antmaze, we still set $\alpha = 1.0$ for the sake of hyperparameter consistency. By keeping $\alpha$ setting constant, we vary $\beta$ and find that OSD-CQL is not very sensitive to changes in $\beta$.

Table 3: Comparison between OSD-BC and other Weighted BC methods

| Method Type | Weighted BC | | | |
|---|---|---|---|---|
| D4RL Task | OptiDICE | PRO-RL | AWR | osd-BC |
| maze2d-u | **123.7** | 8.0 | 1.0 | $123.3 \pm 23$ |
| maze2d-m | **123.8** | 16.3 | 7.6 | $81.0 \pm 9.4$ |
| maze2d-l | 113.3 | 0.1 | 23.7 | $\mathbf{154.0 \pm 21.4}$ |
| maze2d-total | **360.8** | 24.4 | 32.3 | 358.3 |
| hopper-r | 31.3 | 5.1 | 10.2 | $\mathbf{31.5 \pm 0.3}$ |
| hopper-m | 58.3 | 38.1 | 35.9 | $\mathbf{59.7 \pm 3.1}$ |
| hopper-m-r | 27.7 | 32.3 | 28.4 | $\mathbf{35.8 \pm 3.9}$ |
| hopper-m-e | 66.4 | 55.4 | 27.1 | $\mathbf{95.3 \pm 8.0}$ |
| halfcheetah-r | **8.2** | 2.5 | 2.5 | $5.1 \pm 1.3$ |
| halfcheetah-m | 42.3 | 2.5 | 37.4 | $\mathbf{42.5 \pm 0.4}$ |
| halfcheetah-m-r | 38.9 | 2.3 | 40.3 | $\mathbf{39.5 \pm 2.4}$ |
| halfcheetah-m-e | 73.0 | 2.2 | 52.7 | $\mathbf{85.8 \pm 3.8}$ |
| walker2d-r | **8.6** | -0.2 | 1.5 | $5.8 \pm 2.8$ |
| walker2d-m | 53.1 | 1.2 | 17.4 | $\mathbf{73.0 \pm 4.3}$ |
| walker2d-m-r | 54.6 | -0.2 | 15.5 | $\mathbf{56.2 \pm 5.8}$ |
| walker2d-m-e | 88.7 | 0.3 | 53.8 | $\mathbf{107.4 \pm 2.3}$ |
| locomotion total | 551.1 | 141.5 | 322.7 | **637.6** |

It is worth noting that when $\beta = 0$, OSD-CQL performs worse than when $\beta > 0$ in all three domains, which also confirms the role of the regularization term.

## E  COMPARISON BETWEEN OSD-BC AND OTHER WEIGHTED BC BASELINES

We also compare osd-BC against some other weighted BC methods and demonstrate the superiority of our approach. Among the baselines, OptiDICE (Lee et al., 2021) and PRO-RL (Zhan et al., 2022) use the same primal-dual formulation as our method and extract the final policy by density ratio-weighted BC. AWR (Peng et al., 2019) are also weighted BC methods, but use exponential advantage as the importance weights. Upon analyzing Table 3, we observe that osd-BC demonstrates comparable performance to OptiDICE for Maze2d domain, while outperforming other method by a significant margin in the locomotion domain. We attribute this improvement to two factors. Firstly, osd-BC approaches the problem through a simple minimization process, thereby avoiding numerical instability and local convergence issues that may arise with nested optimization methods such as PRO-RL. Secondly, osd-BC learns the near-optimal density ratio with a guarantee of theoretical soundness, which is an advantage not shared by AWR.

## F  COMPARISON BETWEEN OSD-CQL AND OTHER BASELINES

We also evaluate our osd-CQL method against other state-of-the-art (SOTA) methods and several recent reweighing baselines. These baselines consist of EDAC (An et al., 2021), which implements pessimism by considering uncertainty; UWAC (Wu et al., 2021), which reweighs state-action pairs according to uncertainty; IVR Xu et al. (2023), which is an in-sample approach, serves the dual purpose of reweighing data; OPER (Yue et al., 2023), similar to Hong et al., which reweighs data based on both advantage and return; and DM (Hong et al., 2023), which is most relevant to our method, reweighing data using a learned optimal density ratio. The comparison is demonstrated as Table 4, showing that OSD-CQL outperforms other baselines on most datasets. Besides, we also compare OSD-CQL with the reweighing baselines on some mixed datasets in Table 5, the results show that OSD-CQL performs comparable or even better than other baselines on these datasets.

Table 4: Comparison between OSD-CQL and other reweighing methods and SOTA methods

| | OPER-A | EDAC | IVR-SQL | IVR-EQL | DM | UWAC | OSD-CQL |
|---|---|---|---|---|---|---|---|
| maze2d-u | 42.4±5.94 | 25.46±9.93 | 62.43±3.27 | 61.82±20.51 | **172.6±59.62** | 6.1±3.0 | 148.4±33.0 |
| maze2d-m | 19.85±1.91 | 23.22±13.65 | 37.07±2.15 | 4.84±9.36 | 48.77±12.09 | 21.2±2.5 | **102.5±13.0** |
| maze2d-l | 6.25±10.54 | 10.78±18.79 | 56.83±0.82 | 62.64±6.06 | 23.93±36.74 | 6.8±5.6 | **132.2±13.3** |
| hopper-r | 11.75±0.21 | 7.7±0.3 | 0.81±0.04 | 4.52±1.09 | 10.45±3.32 | 2.6±0.1 | **32.6±0.4** |
| hopper-m | 67.4±0.71 | 101.3±0.8 | 60.34±5.09 | 62.75±1.62 | 75.65±11.1 | 49.7±7.4 | **88.6±1.9** |
| hopper-m-r | 99.2±2.55 | 101.5±0.6 | 82.97±11.68 | 64.77±18.70 | 59.64±8.77 | 30.8±13.1 | **100.8±1.0** |
| hopper-m-e | 106.45±0.64 | 88.1±32.3 | 106.75±8.56 | 86.30±4.04 | 57±47.6 | 50.9±7.8 | **109.0±4.7** |
| halfcheetah-r | 24.85±1.48 | **28.4±0.3** | 14.60±0.54 | 15.93±1.11 | 7.97±1.42 | 2.3±0.005 | 27.5±0.1 |
| halfcheetah-m | 49.5±0.28 | **64.2±2.1** | 47.81±0.23 | 49.02±0.04 | 48.21±0.41 | 42.0±0.47 | 59.5±0.45 |
| halfcheetah-m | 46.25±0.07 | **63.3±1.7** | 44.71±0.24 | 44.80±1.10 | 45.41±0.98 | 36.4±4.4 | 51.5±0.2 |
| halfcheetah-m-e | 92±0.57 | 72.2±32.6 | 92.22±2.78 | 82.27±6.52 | 48.62±11.9 | 42.95±0.3 | **93.1±4.7** |
| walker2d-r | 2.55±1.20 | 0.0±0.0 | 0.09±0.12 | 0.78±0.19 | 2.99±3.08 | 2.8±0.2 | **5.6±1.2** |
| walker2d-m | 83.75±1.20 | 89.8±0.4 | 83.20±1.27 | 54.07±6.01 | 69.98±4.03 | 78.3±2.8 | **83.3±1.0** |
| walker2d-m-r | 80.05±11.95 | 81.7±0.1 | 70.86±8.63 | 24.81±10.78 | 75.41±9.02 | 25.5±7.1 | **86.3±8.2** |
| walker2d-m-e | 110.05±0.07 | **113.9±0.4** | 110.92±0.40 | 108.46±3.74 | 100.3±8.28 | 107.16±2.8 | 110.5±0.1 |

Table 5: Comparison between OSD-CQL and other reweighing methods on mixed datasets

| | Uniform | AW | RW | IVR-SQL | IVR-EQL | OPER | DM | OSD-CQL |
|---|---|---|---|---|---|---|---|---|
| hopper-r-m-0.5-v2 | 64.0±1.9 | 60.65±0.19 | 60.65±0.19 | 65.31±0.96 | 4.49±1.82 | 58.8±0.4 | 7.18±9.32 | **75.7±3.9** |
| hopper-r-e-0.5-v2 | 69.6±37.6 | 73.03±2.41 | 73.03±2.41 | **104.46±3.02** | 2.38±0.44 | 84.1±6.8 | 56.05±63.6 | 74.9±9.9 |
| halfcheetah-r-m-0.5-v2 | 49.7±0.8 | 49.1±0.42 | 49.1±0.42 | 45.11±0.47 | 47.04±0.25 | 49.6±0.1 | 48.04±0.07 | **55.3±0.3** |
| halfcheetah-r-e-0.5-v2 | 52.9±2.5 | 65.89±8.58 | 65.89±8.58 | 82.86±3.73 | **86.43±6.18** | 64.2±0.3 | 28.44±3.06 | 69.8±5.1 |

# G    IMPLEMENTATION DETAILS FOR BASELINES

**AW, RW and top10%.**   The BC-related baselines are developed using the official implementation of (Hong et al.). For CQL-based and TD3BC-based baselines, we re-implement top10%, AW and RW strategies on the same codebase (Sun, 2023) to achieve fair comparison. All parameters related to CQL are kept consistent with OSD-CQL. Besides, we adopt the same implementation of (Hong et al.) to implement top10%, AW and RW strategies on top of `https://github.com/young-geng/CQL` for Antmaze domain, to make a fair comparison for this domain. For AW and RW, we perform a search on the hyperparameter $\lambda$ from the set $\{1.0, 0.1, 0.01\}$, and determine that the optimal $\lambda$ is 0.1 so we present the corresponding results for this value.

**EDAC, UWAC and IVR.**   Due to the independence of these baselines from our approach, we reproduce the results using their respective official codes and use the default hyperparameter settings provided in their official codes.

**OPER-A.**   OPER can be combined with different algorithms as a plugin. In order to compare with OSD-CQL, we choose OPER-CQL for comparison. It includes two modes: OPER-A and OPER-R. Since OPER-R mode is similar to RW, and the paper shows that OPER-A has better performance, we choose OPER-A mode for comparison. Specifically, in phase one, we use the official source code of OPER to generate weights. In phase two, we used the weights generated in phase one as sampling weights to reweigh CQL based on the (Sun, 2023). All parameters related to CQL are kept consistent with OSD-CQL.

**DW.**   DW can also be combined with different algorithms as a plugin so we choose DW-CQL for comparison. We re-implemented DW-CQL based on (Sun, 2023), but the performance was not as good as the official source code reproduction. Therefore, we choose to showcase the effect of reproducing the official source code. At the same time, to ensure a fair comparison, we keep the key parameters of DW-CQL and OSD-CQL consistent, including conservative weight, learning rate, network structure, etc.

# H    OBSERVATION STUDY OF THE LEARNED DENSITY RATIO

Before presenting the results, we first visualize the learned density ratio to see if it has learned useful information and revealed any underlying patterns in the datasets. To this end, we plot the histograms of $\widehat{w}_\alpha$ for each dataset. Take "hopper" task in Figure **??** as an example, it can be observed that (i) OSD-DICE does assign distinguishable weights to different samples, and these weights are all distributed within a reasonable range. (ii), the weights for "medium", "medium-replay" and "medium-

expert" datasets are relatively uniformly distributed, while those for "random" dataset are more exclusively focused on $1.0$. This implies that the samples are more distinguishable for "medium", "medium-replay" and "medium-expert", and almost equally bad for "random". (iii) Compared to "medium" and "medium-replay", "medium-expert" has more samples with weights close to $0$ and more samples with weights greater than $4$. This phenomenon aligns with the composition method of "medium-expert", which is a mixture of medium data and expert data. Similar patterns can also be observed for "halfcheetah" and "walker2d" tasks in Appendix H, showing that OSD-DICE can indeed learn the importance levels of different samples in a dataset.

The histograms of the learned density ratio for "halfcheetah" and "walker2d" tasks are shown in Figure 3

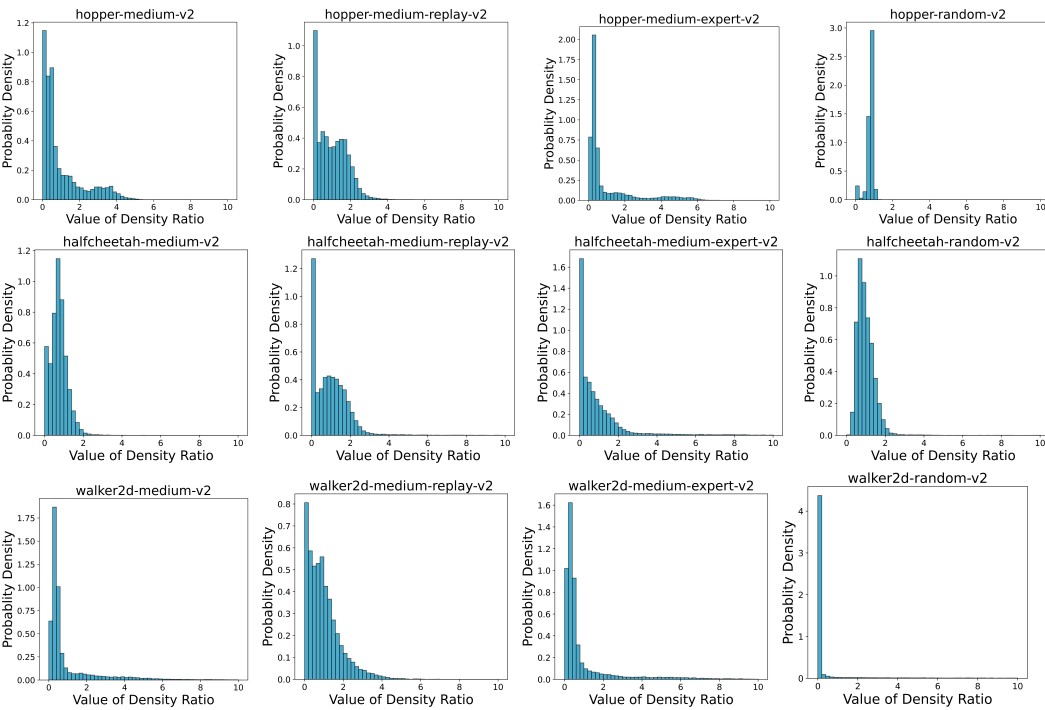

Figure 3: Histograms of the learned density ratio for Halfcheetah tasks and Walker2d tasks. The x-axis represents the value of density ratio, and the y-axis represents the probability density.

