# OpenReview forum: "Enhancing Offline Reinforcement Learning with an Optimal Supported Dataset"
_ICLR.cc/2024/Conference — Submitted to ICLR 2024_

### Official Review · Reviewer_ZUfb · 2023-10-25

**Soundness:** 2 fair
**Presentation:** 2 fair
**Contribution:** 2 fair
**Rating:** 3
**Confidence:** 2

**Summary:**

The work follows the weight approach in offline RL, proposes a new algorithm to find the optimal weight, and further uses the weight to re-weight the dataset, and find the optimal policy.

**Strengths:**

The approach proposed is shown to find a well-behaved policy, with theoretical proofs. The experiment results also verify the capability to find a goo policy.

**Weaknesses:**

1. The writing of this paper should be improved. There are some of the notations that are not pre-defined, which make it hard to read this paper. For example, what is $\mathcal{V}$ in Assumption 1? What is $\mathcal{D_m}$ in page 4? Why does the learner have access to this additional dataset to learn $\hat{P}$? What is $d^*_\alpha$ in (4)? These undefined notations make the paper difficult to understand.
2. There are too many assumptions made to imply the results, which are hard to verify. Also, some justifications should be made regrading these assumptions. For example, how do you verify Assumption 5? In Thm6, some additional assumptions are made. How do you justify those?
3. The sample complexity in Remark 9 is $O(\epsilon^{-4})$, which is much greater than the previous typical results for offline RL, i.e., $O(\epsilon{-2})$. Why is the result worse than the previous ones?
4. The idea of using weight technique is not new. The importance weight technique has been used in RL for a long time, even in the offline RL setting. What is the novelty of this work?

**Questions:**

See above.

---

> ### Author Response · Authors · 2023-11-18
> **Response to Reviewer ZUfb**
>
> We would like to express our heartfelt gratitude for the priceless feedback you have given us. We highly value your viewpoints and believe they will greatly improve our paper. We will carefully address all of your specific questions in our forthcoming response. To make it easier for you to identify the changes, we have highlighted them in blue in our revised version.
>
> **Q: About the symbols**
>
> A: We apologize for not providing you with a clearer understanding of these symbols. $V$ is the optimization space for the neural network $\nu$ that we are studying, which is a common assumption in this type of problem, as described in [1]. The dataset $D_m$ is also a common assumption, as referenced in [2]. $d^{*}_{\alpha}$ represents the optimal support of the data distribution, and we have highlighted its definition in the revised manuscript.
>
> **Q: About Assumption 5 and assumptions in Theorem 6**
>
> A: Assumption 5 is a technical assumption to ensure the continuity of $\hat L_\alpha (\nu)$ for all $\nu \in V$.  It is a constraint on the optimization space $V$ for $\nu$.  Remark 3 associated with Assumption 5 implies  that some desired $V$ can be as follows:  if the advantage  $e_{\nu_\alpha^*}$ induced by the optimal distribution $d_\alpha^*$ is not equal to $\alpha f'(0) $ in the region not visited by  $d_\alpha^*$,  then a neighborhood of the optimal $\nu_\alpha^*$ can be chosen as the desired $V$.   This assumption is not difficult to verify and does not impose strong assumption to the practical problems.
>
>  Theorem 6 does not introduce any new assumptions, but instead requires that the optimal distribution $d^*_0$ when $\alpha=0$ satisfies the common concentrability assumption， Assumptinon 4. Other specific choices regarding $\alpha$ and $f$ are both easily satisfied.
>
>
> **Q: About the sample complexity $O(\epsilon^{-4})$**
>
> A: We acknowledge that we have not proven the sample complexity to be optimal for OSD, as we aimed for practicality. For offline RL methods based on linear programming, although there are some works that can prove the sample complexity to be optimal, they either introduce multiple optimization objectives[2], leading to algorithm instability in practice, or impose constraints that are difficult to apply in real-world scenarios [3].  Therefore, the practicality of OSD is not possessed by  these methods.
>
> However, one of our future goals is to delve into practical algorithms that offer optimal sample complexity.
>
>
> **Q: About the novelty of our method**
>
> A: There does exist some work which  utilize a reweighing strategy to enhance offline RL methods.  The more common practice is to adopt exponentiated advantage or return estimates  as  importance weights, with different methods of advantage estimation [4-6]. While simple and intuitive, these methods do not have a theoretical guarantee of obtaining the near-optimal supported policy. On the contrary, we utilize the optimal density ratio as the importance weight, which is totally different and more theoretically justified. The most pertinent study to our research is [7] conducted during the same period. This study also suggests utilizing optimal density ratios as weights. However, it is worth mentioning that [7] primarily focuses on the scenario where $\gamma=1$. Consequently, they propose a different algorithm to estimate the optimal density ratio, but no evidence is provided to support its optimality. Our method targets the more common case of $\gamma<1$ and theoretically guarantees learning the near-optimal density ratio. Additionally, when combined with typical offline RL algorithms, it can achieve near-optimal supported policies, which is not possessed by the aforementioned works.
>
> [1] Offline reinforcement learning with realizability and single-policy concentrability
>
> [2] Optimal conservative offline rl with general function approximation via augmented lagrangian
>
> [3] Revisiting the Linear-Programming Framework for Offline RL with General Function Approximation
>
> [4] Harnessing Mixed Offline Reinforcement Learning Datasets via Trajectory Weighting
>
> [5] Offline Prioritized Experience Replay
>
> [6] Exponentially Weighted Imitation Learning for Batched Historical Data
>
> [7] Beyond uniform sampling: Offline reinforcement learning with imbalanced dataset

---

### Official Review · Reviewer_MfcJ · 2023-10-29

**Soundness:** 2 fair
**Presentation:** 1 poor
**Contribution:** 3 good
**Rating:** 5
**Confidence:** 4

**Summary:**

This paper deals with the distribution shift problem in offline reinforcement learning through the lens of optimal supported dataset.

Classical methods rely on behavior regularization to constrain the learned policy close to the behavior policy induced by the offline dataset. Such methods have limitations when the behavior policy is sub-optimal. The paper proposes to overcome this drawback by constructing an optimal supported dataset, which can then be used for offline learning via classical regularization-based methods.

The paper starts from the linear programming formulation for MDPs, with the density ratio being a variable. After learning the optimal density ratio, the original dataset can be reweighted. The problem is that a minimax problem is computationally hard to solve. Thus this paper introduces a relaxation and reduces the problem into a single minimization problem.

Furthermore, optimality guarantee is given thanks to the MLE of the transition model and convexity of the objective function brought by an added squared regularization term. Experimental results on RL benchmark D4RL confirms the efficacy of the proposed method.

**Strengths:**

**Significance**: this paper studies the distribution shift problem in offline reinforcement learning, which is an important problem. Furthermore, this paper looks at this problem from the lens of an optimal behavioral dataset, which seems an interesting angle.

**Originality**: this paper contains some original ideas.

**Weaknesses:**

1. **Presentation**:

$(i)$ In LaTex, please use \citet and \citep properly.

$(ii)$ there are some questions not throughly addressed in the paper, which left me very interested. I leave them to the **Questions** session.

2. **Clarity**:

$(i)$ the writing of this paper has a huge room for improvement. It is a bit hard to read. I suggest the authors use tools like ChatGPT to refine the wording. Furthermore, the overall presentation requires a major refinement. Equations like (3) are hard for the readers to interpret if it is just put there without proper elaboration. For example, one possible way is to split eq. 3 into several terms with a notation for each term, and then explain the motivation for each term separately.

$(ii)$ I was a bit confused by the definition of $f$ when I first read the paper. Is it a function class (from Alg.1 line 1) or f-divergence (eq.1)? Things like this need to be defined more clearly.

**Overall**, I think a major modification is necessary for the current manuscript.

**Questions:**

1. The paper claimed that it has computational advantage compared to previous works. However, it seems that solving eq2 requires computation of quantities of MLE. How can this be efficient in cases other than tabular MDPs?

2. Please elaborate on why the added regularization term does not change the optimality. Is bring convexity to the objective function the only purpose of the regularization term?

---

> ### Author Response · Authors · 2023-11-18
> **Response to Reviewer MfcJ**
>
> We want to sincerely thank you for the invaluable feedback you have provided us. We greatly appreciate your perspectives and are confident that they will contribute to the enhancement of our paper. We will systematically address all of your specific inquiries in our upcoming response. To facilitate your review of the modifications, we have marked them in blue in our revised version.
>
> **Q: About clarity and presentation**
>
> A: We apologize for the logical leaps and typographical errors in some of our statements. As a result, we have made revisions to the methodology section. On one hand, we have provided a more detailed explanation of the derivation of equation (3) (now equation (5). On the other hand, we have made changes to the parts you mentioned, hoping to alleviate any confusion or doubts you may have.
>
>
> **Q: About the model learning in complex domains**
>
> A: For complex domains, we have chosen to use the single-transition estimator to approximate $e_{\nu}$. This choice is made because the MDPs involved in these tasks are approximately deterministic, which means that the bias issue is not a  concern as discussed in Corollary 3 in [1]. Therefore, we can rely on the single-transition estimator to provide reliable results in this situation. We have previously clarified this in Appendix C and now emphasized it by providing an explanation in section 5.2.
>
> [1] OptiDICE: Offline Policy Optimization via Stationary Distribution Correction Estimation
>
> **Q: About why the regularization term does not change the optimality and its functions**
>
> A: The reason why the regularization term does not change the optimality has been proven in Lemma 4 in Appendix A, please refer to it for more details.
>
> The introduction of the regularized term in the optimization objective, $L_{\alpha}(\nu)$, is crucial as it brings about properties similar to strong convexity. This plays a significant role in ensuring that the solution obtained from optimizing $L_{\alpha}(\nu)$ is an approximate global optimum. On the other hand, if we were to remove the regularization term, the resulting optimization objective $L_{\alpha}(w^{*}_{\nu}, \nu)$ would lack strong convexity, the  objective function near the optimal value may be too flat, making it difficult for us to bound the difference between the learned solution and the optimal solution.

---

> > ### Comment · Reviewer_MfcJ · 2023-11-21
> > **Response to rebuttal**
> >
> > Dear Authors,
> >
> > Thank you very much for responding to my question.
> >
> > I will take the changes into consideration.
> >
> > Best,
> > Anonymous reviewer

---

### Official Review · Reviewer_NpVF · 2023-11-01

**Soundness:** 3 good
**Presentation:** 3 good
**Contribution:** 2 fair
**Rating:** 5
**Confidence:** 3

**Summary:**

This paper aims to address the limitation of suboptimal datasets for offline reinforcement learning within the framework of DIstribution Correction Estimation (DICE) algorithm. Specifically, this paper proposes Optimal Supported Dataset generation via Stationary DICE (OSD-DICE) method to learn the density ratio for the regeneration of near-optimal dataset. Like prior OptiDICE, the proposed method also adopts a single minimization objective, and incorporates two designs (adding strong convexity terms and learning transition model for the objective) for better optimization and sample complexity.

**Strengths:**

1. Sufficient theoretical analysis of the proposed method;
2. The idea of the proposed method is general and can be extended to other similar offline RL methods.

**Weaknesses:**

1. Lack the sufficient comparison with current SOTA offline RL algorithm, such as, EDAC, IQL, RORL, or it's worth considering combing this learned near-optimal dataset with these SOTA algorithms;
2. Compared with original OptiDICE, the main innovations of this paper focus on two design, but it rarely explains and verifies what key problems these designs can solve.
3. The proposed two improvements lacks sufficient novety, adding strong convexity regularization term (last term in e.q.3) is a popular trick in this type of methods.

**Questions:**

1. It's better to supplement more comparison with current SOTA offline RL methods;
2. It's said that, when estimating the Bellman error $e_\nu$, this paper explicitly considers the transition probability of the next state which is believed to bypass the bias issue. Would you please make it more clear why the original single-transition estimation suffers from the bias issue? In my opinion, this estimation is based on Monte Carlo sampling and hence being unbiased but have high variance. Besides, introducing an extra approximate transition probability fucntion $\hat{P}$ doesn't seems reliable to address the bias issue - how can you ensure the accuracy of the model?

---

> ### Author Response · Authors · 2023-11-18
> **Response to Reviewer NpVF (1)**
>
> We would like to express our heartfelt gratitude for the valuable feedback you have given us. Your insights are highly valued and will undoubtedly help us improve our paper. We will address each of your specific questions in a systematic manner in our next response. In order to make it easier for you to see the changes we have made, we have highlighted them in blue in our revised version.
>
> **Q： More comparison with SOTA methods**
>
> A: Thank you very much for your advice. We have added  EDAC, and some recent reweighing methods as new baselines, as shown in Appendix F, Table 4. The results indicate that OSD method outperforms these baselines in most of the datasets.
>
> **Q：  About the explanations and verifications of two key designs**
>
> A: We apologize for not being able to clearly demonstrate the roles and effects of the two core designs. Their respective functionalities are explained as follows：
>
> * The objective of model learning is to mitigates the bias issue that arises in OptiDICE. This is achieved by ensuring that the empirical objective and the expected objective can be closely controlled, especially when the sample size $N$ is large. Additionally, the error associated with this treatment depends solely on the size of $|\mathcal{P}|$ and $N$.
>
> * The introduction of the regularized term in the optimization objective, $L_{\alpha}(\nu)$, is crucial as it brings about properties similar to strong convexity. This plays a significant role in ensuring that the solution obtained from optimizing $L_{\alpha}(\nu)$ is an approximate *global* optimum. On the other hand, if we were to remove the regularization term, the resulting optimization objective $L_{\alpha}(w^{*}_{\nu}, \nu)$ would lack strong convexity, the behavior of the objective function near the optimal value may be too flat, making it difficult  to bound the difference between the learned solution and the optimal solution, potentially leading to the presence of alternative  optima.
>
> In order to provide a clearer explanation of their functions, we have explained the motivation and effects of these two core designs in Section 4.2.
>
> Regarding their practical effects in experimental settings, we have already shown their orthogonal functionality in tabular MDPs in Section 5.1. As for the experimental utility in complex tasks, the verification has been deferred in Appendix D and E due to space limitations.  To be more highlighted, we have added a paragraph in Section 5.2 to discuss the contribution of the regularization term.
>
> **Q: About the novety of two key designs**
>
> * Firstly, both of these designs play a crucial role in theory. OptiDICE, by utilizing a single transition estimator, leads to a biased empirical objective of the original objective. To tackle this issue, we introduce an approximated transition estimator, which allows us to  control the error between  the empirical objective and the expected objective. The error associated with this approach depends solely on the size of $|\mathcal{P}|$ and $N$. However, even with this modification, solving the new empirical objective may not guarantee the attainment of the global optimal solution. To overcome this limitation, we incorporate a regularization term that does not impact the solution but enhances the convexity of the optimization objective. This regularization term ensures that the learned solution is indeed the optimal solution. Therefore, although the implementation of these two designs is not complex, they are necessary conditions to ensure the optimality of the algorithm.
>
> * Secondly, the proof process is not straightforward. On one hand, we need to demonstrate that the introduction of the regularization term does not change the true optimal solution. On the other hand, due to the complexity introduced by equation (5), it does not strictly adhere to convexity principles, making it impossible to directly apply the properties of strong convex functions. Consequently, we must thoroughly analyze the characteristics of equation (5) in various scenarios, particularly in the vicinity of the optimal solution, to acquire the assurance of optimality.
>
> * Finally, our framework goes beyond the mere solution of the optimization problem $\min_{\nu}\widehat{L}_{\alpha}(\alpha)$ by enabling the incorporation of the learned optimal weights with BC or behavior regularization-based algorithms. This integration enhances the versatility and effectiveness of our framework. Moreover, we offer a theoretical guarantee, as demonstrated by Theorem 10 and Remark 11, that the resulting policy derived from our framework will not be inferior to the optimal policy supported by the dataset. This novel contribution represents a significant advancement that has not been previously explored.

---

> ### Author Response · Authors · 2023-11-18
> **Response to Reviewer NpVF (2)**
>
> **Q:Explanation of bias issue induced in single-transiton estimator of $e_{\nu}$.**
>
> We apologize for not being able to explain the bias issue in the main text more clearly. The background of this question is as follows.
>
> The original optimization objective  is  $L_\alpha \left(w_\nu^*, \nu\right)$  which is defined in (4) in the revision.
> OptiDICE utilizes a single-transition estimation $\tilde e_\nu = r(s,a) + \gamma \nu(s') - \nu(s)$  to approximate  $e_\nu$ in (4),
> which thus solves an empirical objective $\hat L_\alpha(w_\nu^*,\nu)$.  However, due to the non-linearity of $(f')^-1$ and the double-sample problem, it has been proven by Jenson Inequality in Corollary 3 of [1] that $\hat L_\alpha(w_\nu^*,\nu) \geq L_\alpha\left(w_\nu^*, \nu\right)$, where equality holds when the MDP is deterministic. Therefore, a bias arises for OptiDICE.
>
> However, in complex domains, we actually still adopt the single-transition estimator to approximate $e_\nu$, as the MDPs are approximately deterministic on these tasks, and the single-transition estimator is simple and reliable in this situation. This has been explained in Section 5.2.
>
> [1] OptiDICE: Offline Policy Optimization via Stationary Distribution Correction Estimation

---

### Official Review · Reviewer_yzkc · 2023-11-01

**Soundness:** 3 good
**Presentation:** 2 fair
**Contribution:** 2 fair
**Rating:** 6
**Confidence:** 4

**Summary:**

The contribution of this paper is two-fold:
- Proposes an algorithm called OSD-DICE that learns a model and adopt a squared regularization for w for better convexity. There is some theoretical backups for the regularization. In finite MDP experiment, the improvement over OptiDICE is shown.
- Proposes a dataset resampling algorithm based on BC and CQL is introduced using the weight optimized from OSD-DICE, and shows the improvement over baseline offline RL algorithms in D4RL environments.

**Strengths:**

- Strong theoretical analyses on OSD-DICE.
- Shows much improvement in experiments. OSD-DICE shows improvement over OptiDICE with its corrections. In D4RL experiment, it is shown that CQL can be benefit from OSD-DICE based importance sampling.

**Weaknesses:**

- OSD-DICE requires additional learning of a model compared to OptiDICE, and it is not certain whether the performance improvement over OptiDICE is worth learning a model in complex domains. For finite domains, while the squared regularization seems to improve a lot, but using a model for unbiased estimator does not seem to improve much. It is counterintuitive as model learning makes the estimator unbiased, while squared regularization does not change the optimal solution (in theory).
- While OSD-DICE seems to be an improved version of OptiDICE, in the importanced sampled RL part, the proposed algorithms are not compared against OptiDICE-based algorithms. This makes two contributions of the paper to feel very separated.

**Questions:**

- As far as I understood, the additional regularization allows us to make the problem well defined even for 0 like $\alpha$, as it offers additional convexity. But shouldn't the optimal solution be the same? What's the main reason that the adoption of regularization can improve from OptiDICE in finite MDP? we should be able to optimize to the optimal solution in finite MDPs, so the solution should stay the same?
- Among two contributions of this paper (1. OSD-DICE against OptiDICE, 2. OSD-DICE based importance sampled offline RL), the second one seems to be very similar to that of [1]. Does the proposed method perform better than [1]?

[1] Beyond Uniform Sampling: Offline Reinforcement Learning with Imbalanced Datasets

---

> ### Author Response · Authors · 2023-11-18
> **Response to Reviewer yzkc**
>
> We would like to express our heartfelt gratitude for offering us valuable feedback. Your thoughtful comments are highly valued as they will undoubtedly help improve our paper. In our upcoming response, we will carefully address each of your specific questions. To ensure clarity, we have indicated all the changes and additions made in our revised version by highlighting them in blue.
>
> **Q: About the comparison with OptiDICE**
>
> A:  We sincerely apologize for the oversight of not including a direct comparison between OSD-BC and OptiDICE for complex tasks in the main text. Due to space limitations, we had to defer the comparison to Appendix D and E.
>
> * In Appendix E Table 3, we have provided an explicit comparison between OSD-BC and OptiDICE. The results indicate that they perform comparably in the maze2d domain, but OSD-BC has a remarkable advantage over OptiDICE in the Locomotion domain.
>
> * Furthermore, we conducted ablation experiments on the regularization coefficient $\beta$, as shown in Appendix D Figure 2(b). The variant with $\beta=0$ corresponds exactly to OptiDICE+CQL. It is worth noting that OSD-CQL consistently outperforms OptiDICE+CQL across all three tasks. This highlights the advantage of the OSD method over OptiDICE when incorporated with behavior regularization-based offline methods.
>
> To provide a more comprehensive comparison between OSD and OptiDICE in complex tasks, we have added a paragraph in Section 5.2 of the main text to discuss the comparison between the two methods.
>
> **Q: OSD-DICE requires additional learning of a model compared to OptiDICE, and it is not certain whether the performance improvement over OptiDICE is worth learning a model in complex domains.**
>
> A:  For complex domains, we have chosen to use the single-transition estimator to approximate $e_{\nu}$. This decision is based on the fact that the MDPs involved in these tasks are approximately deterministic, which means that the bias issue is not a major concern as discussed in Corollary 3 in [2]. Therefore, we can rely on the single-transition estimator to provide reliable results in this situation. We explained this point previously in the Appendix C, and now we have highlighted it by providing an explanation in section 5.2.
>
> **Q: For finite domains, while the squared regularization seems to improve a lot, but using a model for unbiased estimator does not seem to improve much. It is counterintuitive as model learning makes the estimator unbiased, while squared regularization does not change the optimal solution (in theory).**
>
> A： For a finite MDP, the single transition estimator $e_{\nu}$ is essentially replacing the true transition $P$ with the empirical estimator $\tilde{P}$ obtained from the dataset. While there is some bias in $\tilde{e}_{\nu}$ and the induced objective, the bias is generally small when the dataset size is sufficiently large. This observation helps explain why the performance of model learning is often not that bad compared to not performing model learning.
>
> However, it is important to note that the inclusion of model learning is theoretically necessary. This is because model learning allows us to bound the bias, and the upper bound is solely dependent on the size of the candidate function class $|\mathcal{P}|$ and the size of the dataset $N$.
>
> **Q： About the reason to use additional regularization term**
>
> A:  We propose the addition of a regularization term to enhance the convexity of the optimization objective (5), aiming to transform it into  an approximately strongly convex function. This regularization term plays a crucial role in guaranteeing that the solution obtained from optimizing (5) is an approximate *global* optimum. Conversely, if we were to eliminate the regularization term, the resulting optimization objective would lack strong convexity, the behavior of the objective function near the optimal solution may be too flat, making it difficult for us to bound the difference between the learned solution and the optimal solution, potentially leading to an alternative optima. To clarify the intention behind this regularization term, we have included a description Section 4.2.
>
> **Q: Comparison with [1]**
>
> A： Thank you very much for pointing out this relevant work. We have compared this work in the main text and supplemented experiments comparing it. The experiments show that the OSD method still has advantages compared to it.
>
> [1] Beyond Uniform Sampling: Offline Reinforcement Learning with Imbalanced Datasets
>
> [2] OptiDICE: Offline Policy Optimization via Stationary Distribution Correction Estimation

---

### Official Review · Reviewer_tQGo · 2023-11-01

**Soundness:** 3 good
**Presentation:** 2 fair
**Contribution:** 3 good
**Rating:** 8
**Confidence:** 2

**Summary:**

The paper introduces a new approach to improve offline RL by addressing the distributional shift and value overestimation issues that degrade policy performance. The core contribution is Optimal Supported Dataset generation via Stationary DIstribution Correction Estimation (OSD-DICE), a method that formulates the generation of an optimal supported dataset through a refined primal-dual linear programming process. This approach simplifies the optimization to a single minimization objective, avoiding the instability of traditional methods, and is theoretically robust, offering polynomial sample complexity with general function approximation and single-policy concentrability.

To validate their approach, the authors incorporate the generated dataset into two behavior regularization methods—Behavior Cloning (BC) and Conservative Q-Learning (CQL)—creating osd-BC and osd-CQL. They demonstrate that these methods achieve safe policy improvement and enhanced performance on D4RL benchmarks. The experiments confirm the efficacy of OSD-DICE and suggest that using an optimal supported dataset can substantially benefit offline RL tasks.

**Strengths:**

- Theoretical analysis: The OSD-DICE framework presents a theoretically sound approach, advancing the field with its single minimization objective that resolves the complexity and instability issues found in traditional primal-dual optimization methods for optimal support dataset methods.

- Empirical results: The paper supports its theoretical claims with robust empirical evidence, demonstrating significant performance improvements on well-established D4RL benchmarks, which suggests the method's practical effectiveness in offline RL tasks.

**Weaknesses:**

The presentation of Equation (3) lacks an intuitive explanation. The paper would benefit from a clearer exposition of this optimization problem, helping readers better grasp its significance within the OSD-DICE framework and enhancing overall accessibility.

**Questions:**

N/A

---

> ### Author Response · Authors · 2023-11-18
> **Response to Reviewer tQGo**
>
> Thank you very much for your recognition of our work and your constructive suggestions. Our response is as follows.
>
> **Q: About explanation of Equation (3)**
>
> A:  We are sorry that we could not make the source of equation (3) clearer. We have provided the background and context of equation (3) (now equation (5)) in Section 4.1 of the main text, highlighted in blue. We hope this helps you better understand the rationale and consequences of this optimization objective.

---

### Official Review · Reviewer_v4Ur · 2023-11-02

**Soundness:** 2 fair
**Presentation:** 3 good
**Contribution:** 2 fair
**Rating:** 5
**Confidence:** 3

**Summary:**

The author introduces a method for reweighting datasets to optimize the behavior policy in behavior-regularized offline Reinforcement Learning (RL) for enhanced performance. The effectiveness of behavior-regularized offline RL algorithms is contingent on the behavior policy that accumulates the dataset. A prevalence of low-return trajectories in the behavior policy’s collected data results in diminished performance of the offline RL algorithms. The proposed method modifies the behavior policy by adjusting the dataset weights, employing the density importance correction estimation (DiCE) technique to optimize these weights, thereby enhancing the performance of the behavior policy. Experimental outcomes indicate performance enhancements in several D4RL offline RL datasets.

**Strengths:**

- The proposed method is well-motivated, and using DiCE for weighting data is new.
- The theoretical results are encouraged.

**Weaknesses:**

- Lack of baselines and related works discussion. Re-weighting offline RL training objectives or optimizing the dataset has been studied in prior works [1, 2, 3, 4]. However, the author didn't discuss and compare with them. Without comparing with these works, it would be difficult to answer the significance of this work.
- Implementation details of the baselines are not presented. The results of Table 1 look pretty similar to the official codebase of AW/RW. However, it should be noted that the official codebase of AW/RW uses offline RL hyperparameters (e.g., the regularization weight of CQL) different than Sun 2023. It's necessary to set those offline RL's hyperparameters to be the same for fair comparison. It'd be great if the author can upload their baseline implementation and scripts to reproduce experiments.
- Hyperparameter search results of baselines are not presented even though the author claims that the baselines' hyperparameters are also optimized.
- The baseline scores in Table 1 don't have a standard deviation.
- Performance improvement is limited, except for maze2d. From Table 1, OSD doesn't show a significant performance gain in MuJoCo tasks. Checking Lee et al. (2021), we see that their method also performs the best in maze2d but underperforms or match the baselines in others. But, the author doesn't compare Lee et al. (2021) in Table 1, which makes the contribution of OSD-RL unclear.
- OSD-RL requires training two additional models, which increases the run time. The author should compare the run time (wallclock time) with the baselines. For this method to be impactful, the increased runtime should be proportional to their performance gain.

[1] Wu, Yue, et al. "Uncertainty weighted actor-critic for offline reinforcement learning." arXiv preprint arXiv:2105.08140 (2021).

[2] Chen, Xinyue, et al. "Bail: Best-action imitation learning for batch deep reinforcement learning." Advances in Neural Information Processing Systems 33 (2020): 18353-18363.

[3] Xu H, Jiang L, Li J, et al. Offline rl with no ood actions: In-sample learning via implicit value regularization[J]. arXiv preprint arXiv:2303.15810, 2023.

[4] Yue, Yang, et al. "Offline Prioritized Experience Replay." arXiv preprint arXiv:2306.05412 (2023).

**Questions:**

- What's the difference between OptiDiCE + SquareReg and OSD-DiCE?
- It would strengthen the paper's impact if the author could show the performance in all D4RL datasets, Atari domains, and potentially the imbalanced datasets proposed in AW/RW paper.
- I encourage the author to upload the source code and the scripts to reproduce the experiments as D4RL experiments shouldn't take long time to run.

---

> ### Author Response · Authors · 2023-11-18
> **Response to Reviewer v4Ur**
>
> Thank you sincerely for providing us with constructive feedback. We greatly appreciate your insights as they will undoubtedly contribute to the enhancement of our paper. In response to your specific inquiries, we will systematically address each one in our subsequent response. To facilitate clarity, we have highlighted all  modifications and additions in blue in our revision.
>
> **Q: About the lack of baselines and related works discussion**
>
> A: Thank you for your suggestion. We have discussed and compared the literature mentioned by you in Section 2 of the main text. Additionally, we have included the results of these baselines in Appendix F, Table 4-5. The results indicate that OSD-CQL outperforms these baselines in most of the datasets.
>
> **Q: About the implementation details of the baselines.**
>
> A: We have added implementation details about the baseline in the revised paper, please refer to Appendix G. It is worth mentioning that both the TD3BC-based baselines and the CQL-based baselines are indeed implemented based on [Sun], and the hyperparameters are consistent with osd-TD3BC and osd-CQL, which ensures a fair comparison. We have uploaded our  source code package, which includes our algorithms, baseline code, and execution scripts.
>
>
> **Q: About the hyperparameter search results for baselines.**
>
> A: We have added the method and results for selecting hyperparameters of the baselines. Please refer to Appendix G.
>
> **Q: About the lack of  standard deviation for baseline scores.**
>
> A: We apologize for not including the std in Table 1 in the main text due to space limitations. For the newly added experiments, we present the std, as shown in Table 4-5 in Appendix F.
>
>
> **Q: About the comparison with OptiDICE**
>
> A: We apologize for not presenting the comparison between OSD-BC and OptiDICE for complex tasks  in the main text. Actually, the comparison has been deferred in Appendix D and E due to space limitations.
> - Firstly, the explicit comparison between OSD-BC and OptiDICE is put in Appendix E Table 3.  The results show that they perform comparable in maze2d domain, but OSD-BC has a significant advantage over OptiDICE in Locomotion domain.
> - Additionally, we conducted ablation experiments on the regularization coefficient $\beta$, as shown in Appendix D Figure 2(b). The variant with $\beta=0$ corresponds exactly to OptiDICE+CQL. It can be observed that OSD-CQL consistently outperforms OptiDICE+CQL across all three tasks. which highlights the advantage of  OSD method over OptiDICE when imcorperated with behavior regularization based offline methods.
> - In order to provide a clearer and more explicit comparison between  OSD  and OptiDICE in complex tasks, we have added a paragraph in Section 5.2 of the main text to discuss the comparison between the two.
>
> **Q: About the siginificance of OSD-RL**
>
> * Firstly, as shown in Table 1 in the main text, OSD consistently brings significant improvements to the baselines without any reweighing treatment, take CQL based experiments as an example,  the total performance gain is about 300+ points on maze2d domain, 100+ points on locomotion domain and 100+ points on Antmaze domain.
> * Secondly, from Table 1 again, OSD has remarkable advantages over other mainstream reweighing methods, including top10%, AW and RW, espeically incorperated with behavior regularization based methods. This further validates the effectiveness and universality of OSD.
> * At last, we also compare OSD-CQL with additional baselines as the reviewer suggested. The results show that OSD-CQL still has remarkable advantages compared to these newly added baselines.
>
> **Q: About the difference between OptiDiCE + SquareReg and OSD-DiCE**
>
> A： We apologize for not providing a more detailed explanation about the differences between OSD-DICE and OptiDiCE + SquareReg in the main text. Both OSD-DICE and OptiDiCE + SquareReg use squared regularization in the optimization objective. The difference lies in the fact that OSD-DICE utilizes an approximator $\widehat{P}$ in a realizable function class $\mathcal{P}$ to approximate  the advantage function $e_{\nu}$ and OptiDiCE + SquareReg uses the empirical $\tilde{P}$ to approximate $e_{\nu}$. Detailed construction method for $\tilde{P}$ is put in Appendix B. The comparison between the two is to verify the effect  of approximate advantage in the tabular case.
>
> **Q: About exmperiments for more tasks**
>
> A: Thank you for your suggestion. We have added experimental results on  some mixed datasets. See Table 5 in Appendix F for more details.

---

> > ### Comment · Reviewer_v4Ur · 2023-12-05
> >
> > Thanks for the responses and addressing my questions. I've raised the score to a 5. I cannot raise the score higher because the added experiments are not comprehensive and only conducted in part of the environment.

---

### Meta-Review · Area_Chair_bwVz · 2023-12-05

**Metareview:**

This work contributes a novel method for data regularization of training offline RL methods with robust experiments and theoretical analysis. While there is disagreement in the reviewers, the authors have gone to great lengths to enhance their theoretical analysis to address all concerns. However, I will not overrule the most-probable opinion, which is that it is marginally below the bar for acceptance at this time.

**Justification For Why Not Higher Score:**

The reviewers are not satisfied with the thoroughness of the experimental evaluation and the intuitive justification for why the proposed methodology helps.

**Justification For Why Not Lower Score:**

NA

---

### Decision · Program_Chairs · 2024-01-16

Reject